# Overcoming the Convex Barrier for Simplex Inputs

**Harkirat Singh**
University of Oxford
harkirat@robots.ox.ac.uk

**M. Pawan Kumar**
DeepMind
mpawan@deepmind.com

**Philip H.S. Torr**
University of Oxford
phst@robots.ox.ac.uk

**Krishnamurthy (Dj) Dvijotham**
DeepMind
dvij@google.com

## Abstract

Recent progress in neural network verification has challenged the notion of a *convex barrier*, that is, an inherent weakness in the convex relaxation of the output of a neural network. Specifically, there now exists a tight relaxation for verifying the robustness of a neural network to $\ell_\infty$ input perturbations, as well as efficient primal and dual solvers for the relaxation. Buoyed by this success, we consider the problem of developing similar techniques for verifying robustness to input perturbations within the probability simplex. We prove a somewhat surprising result that, in this case, not only can one design a tight relaxation that overcomes the convex barrier, but the size of the relaxation remains linear in the number of neurons, thereby leading to simpler and more efficient algorithms. We establish the scalability of our overall approach via the specification of $\ell_1$ robustness for CIFAR-10 and MNIST classification, where our approach improves the state of the art verified accuracy by up to $14.4\%$. Furthermore, we establish its accuracy on a novel and highly challenging task of verifying the robustness of a multi-modal (text and image) classifier to arbitrary changes in its textual input.

## 1 Introduction

Verification refers to the challenging computational problem of determining whether a neural network satisfies a given specification. Perhaps the most popular specification is to prove or disprove that a neural network classifier is robust to perturbations of the input that lie within an $\ell_p$ ball [de Palma et al., 2021a, Wong and Kolter, 2018, Zhang et al., 2018, Weng et al., 2018, Mirman et al., 2018, Dvijotham et al., 2018], with impressive state of the art results recently achieved in Wang et al. [2021]. The ability to verify neural networks would open the door to better understanding their nature, and allow us to apply deep learning in safety critical domains where errors can have a large cost. Given the importance of the problem, it is unsurprising that it has attracted considerable atttention from the machine learning and automated verification communities [Katz et al., 2017, Ehlers, 2017].

Early work in verification lead to the notion of a *convex barrier*, that is, a limitation in the tightness of the bounds obtainable by the convex relaxation of the output of a neural network [Salman et al., 2019]. This weakness was seen as a primary reason for the slow convergence of branch-and-bound algorithms for verification, which rely on convex relaxations to compute the bounds, on a large number of specifications on standard datasets. However, recent work has been able to successfully overcome this barrier. Specifically, Anderson et al. [2019] proposed a tight relaxation that precisely defines the convex hull of a composition of a linear function of a vector within an $\ell_\infty$ ball with the ReLU non-linearity. While their relaxation has a large number of constraints (exponential in the size of the vector), they provide an efficient algorithm for identifying the most violated constraint at any infeasible point. This enables the use of efficient cutting plane algorithms to solve the

primal [Anderson et al., 2019], active sets to solve the dual [de Palma et al., 2021a,b], and even approximate linear bound propagation algorithms [Tjandraatmadja et al., 2020].

Buoyed by the possibility of realizing practical verification using tight relaxations, we consider an important specification, namely, robustness to perturbations that lie in a probability simplex. Examples of such a specification include robustness to $\ell_1$ perturbations or to word substitutions [Huang et al., 2019]. Previous approaches for addressing this specification relied on fairly loose relaxations based on interval bound propagation [Huang et al., 2019, Gowal et al., 2018]. To alleviate this deficiency, we derive the convex hull of the composition of a linear transformation of the probability simplex with a convex non-linearity such as ReLU or SoftPlus. Somewhat surprisingly, we show that, unlike the previously considered case of $\ell_\infty$ balls, the probability simplex helps greatly simplify the description of the convex hull. In fact, the number of constraints required is linear in the dimensionality of the simplex. By using a novel technique to propagate the simplex constraints through the hidden layers of the network, we derive a tight relaxation for the output of a network whose size scales linearly with the size of the network. Furthermore, we suitably extend a linear bound propopagation algorithm [Zhang et al., 2018] to solve the tight relaxation efficiently, thereby realizing practical verification over simplex inputs.

We demonstrate the scalability of our approach using the specification of robustness to $\ell_1$ perturbations for MNIST and CIFAR-10 classification. Our method achieves $13.6\%$ higher verified accuracy on MNIST and up to $14.4\%$ higher verified accuracy on CIFAR-10 compared to the state of the art baselines on the same networks given the same computational budget.

To further demonstrate the benefits of our tight relaxation, we consider a novel and highly challenging specification of global robustness in multi-modal classification. Specifically, we consider the Food 101 dataset [Wang et al., 2015], where each sample consists of an image of a food item together with its recipe. Our specification requires a neural network trained to classify the food item to be robust to arbitrary changes in the text of a sample. This specification captures the scenario where the image is carefully curated while the text is crowd-sourced and can therefore be easily manipulated by adversarial actors. While similar specifications have been considered in previous works [Jia et al., 2019, Huang et al., 2019], they focus on substituting a very small subset of words with their synonyms, which leads to simpler verification problems. We show that, on our significantly more difficult global verification task, our method achieves up to 25% higher verified accuracy in the same amount of time as the state of the art baseline.

## 2 Preliminaries

We provide a formal mathematical description of our verification problem, and briefly discuss the existing relaxations in order to contextualise our contributions.

### 2.1 Problem description

We denote the $m$-dimensional probability simplex as $\Delta_m$, that is, $\Delta_m = \{\boldsymbol{x} \in \mathbb{R}^m \mid \boldsymbol{x} \geq 0, \mathbf{1}^\top \boldsymbol{x} \leq 1\}$. As mentioned in the previous section, we are interested in verifying specifications of a given neural network when its input is constrained to lie in a simplex. We focus on networks with a layered architecture to keep notation simple, but our ideas easily extend to any feed-forward network, including residual networks. We consider a neural network with $n$ layers. Each layer is assumed to be composed of two operations: (i) an affine operation (fully connected layer or convolution layer), which we denote by $\mathbb{L}_k(\cdot)$; and (ii) a non-linear activation function, which we denote by $\sigma(\cdot)$. In other words, given its input $\boldsymbol{x}_{k-1} \in \mathbb{R}^{n_{k-1}}$, the $k$-th layer performs the operation $\hat{\boldsymbol{x}}_k = \mathbb{L}_k(\boldsymbol{x}_{k-1}) \in \mathbb{R}^{n_k}$, followed by $\boldsymbol{x}_k = \sigma(\hat{\boldsymbol{x}}_k) \in \mathbb{R}^{n_k}$. While we place no restriction on the linear operation, we make the following assumption on the activation function.

**Assumption 1.** We assume that the activation function $\sigma$ is an element-wise convex function (for example, ReLU or SoftPlus).

**Verification problem:** Using the above notation, the verification problem we solve can be formulated as

$$\min_{\boldsymbol{x}} \quad \Psi(\hat{\boldsymbol{x}}_n) \tag{1a}$$

$$s.t. \quad \hat{\boldsymbol{x}}_k = \mathbb{L}_k\left(\boldsymbol{x}_{k-1}\right), \boldsymbol{x}_k = \sigma(\hat{\boldsymbol{x}}_k) \qquad\qquad k \in [n], \qquad\qquad \text{(1b)}$$

$$\boldsymbol{x}_0 \in \Delta_{n_0}, \qquad\qquad\qquad\qquad\qquad\qquad\qquad\qquad \text{(1c)}$$

where $[n]$ denotes the set $\{1, \cdots, n\}$. The objective function $\Psi$ is a scalar-valued linear function of the output of the final layer of the the network. For example, say we wish to verify that a classification network is not vulnerable to any adversarial attacks. In this case, we define $\Psi$ as the difference between the true logit and a target logit outputted by the network. The sign of the optimum value of problem (1) can be used to either prove or disprove the aforementioned specification.

We illustrate the practical importance of considering simplex inputs using two examples. These examples will also form the basis of our experimental setup.

$\ell_1$ **perturbations:** Consider a network with a continuous valued input $\boldsymbol{x} \in \mathbb{R}^m$. We are interested in verifying the behavior of the network under input perturbations that lie within an $\ell_1$ ball: $\{\boldsymbol{x} \mid \left\|\boldsymbol{x} - \boldsymbol{x}^0\right\|_1 \le \epsilon\}$. This input domain can be reformulated as a simplex as

$$\boldsymbol{x} = \boldsymbol{x}^0 + \epsilon M \boldsymbol{z}, \ \boldsymbol{z} \in \Delta_{2m}, \ \underset{(m \times 2m)}{M} = \begin{pmatrix} 1 & -1 & 0 & 0 & \ldots & 0 & 0 \\ 0 & 0 & 1 & -1 & \ldots & 0 & 0 \\ \vdots & & & & & & \\ 0 & 0 & 0 & 0 & \ldots & 1 & -1 \end{pmatrix}. \qquad \text{(2)}$$

**Bag of words models:** Consider a text classification network that takes text as input and makes predictions based on an embedding of the text. A commonly used embedding is the so-called "bag of words". Here, we first take an embedding for every word in the text (for example, using a precomputed set like GloVe [Pennington et al., 2014] or Word2Vec [Mikolov et al., 2013]). Next, we take the mean of all the word embeddings to obtain the final representation. We denote the word embeddings as a matrix $E \in \mathbb{R}^{d \times v}$, where each $d$-dimensional column represents an embedding of a putative word in a vocabulary of size $v$. Using the above matrix, the text embedding is given by $E\boldsymbol{x}$ where $\boldsymbol{x}$ represents the normalized counts of words from the vocabulary in the text. In other words, $\boldsymbol{x} \in \Delta_v$ assuming that arbitrarily long text with arbitrary numbers of repetitions of each word in the vocabulary are allowed.

## 2.2 Planet and disjunctive relaxations

Ehlers [2017] proposed a convex relaxation for the ReLU activation function, which is commonly referred to as Planet in the literature. The Planet relaxation has been widely used in many verification algorithms [Bunel et al., 2020b, Dvijotham et al., 2018, Bunel et al., 2020a, Lu and Kumar, 2020]. Briefly, it relaxes the sequence of two operations: $\hat{x} = \boldsymbol{w}^T \boldsymbol{x} + b$ where $\boldsymbol{x} \in \mathbb{R}^m$, followed by $y = \text{ReLU}(\hat{x})$. To this end, it utilizes lower and upper bounds on $\hat{x}$. In our case, since $\boldsymbol{x} \in \Delta_m$, we can compute the bounds as $\tilde{\ell} = w_{\min} + b \le \boldsymbol{w}^T \boldsymbol{x} + b \le w_{\max} + b = \tilde{u}$. Here, $w_{\min}$ and $w_{\max}$ denote the minimum and maximum entries of $\boldsymbol{w}$ respectively. Using the bounds on $\hat{x}$, the Planet relaxation for the set $\mathcal{S} = \{y, \boldsymbol{x} \mid y = \text{ReLU}(\boldsymbol{w}^T \boldsymbol{x} + b), \boldsymbol{x} \in \Delta_m\}$ is defined as

$$\mathcal{P}_\Delta : y \ge \boldsymbol{w}^T \boldsymbol{x} + b, \ y \ge 0, \ y \le \frac{\tilde{u}}{\tilde{u} - \tilde{\ell}} \left(\boldsymbol{w}^T \boldsymbol{x} + b - \tilde{\ell}\right), \ \boldsymbol{x} \in \Delta_m. \qquad \text{(3)}$$

In Anderson et al. [2019, 2020], the authors propose a tighter relaxation and characterize the exact convex hull of the set

$$\{(x, \text{relu}\left(\boldsymbol{w}^T \boldsymbol{x} + b\right)) : \ell \le x \le u\}.$$

However, this characterization involves exponentially many inequalities. And efficient algorithms based on it need to resort to cutting plane methods, adding violated inequalities sequentially. Doing so is computationally challenging and requires significant effort to implement in a scalable manner. Further, this does not handle the simplex constraint on the input. Thus using this relaxation would require replacing the simplex constraint with the unit hypercube, a much weaker constraint on the inputs.

Anderson et al. [2020] also propose an ideal formulation for the product of $k$ simplices. Their formulation needs additional variables and still requires cutting planes. However, eliminating the auxiliary variable, as in our case, leads to a concise formulation. Further, our relaxation is derived for convex activation functions, whereas the formulation of Anderson et al. [2020] is limited to the maximum of two affine functions.

# 3 A Concise Convex Relaxation

In this section, we derive our novel tight convex relaxation for problem (1). In order to make the exposition clearer, we assume that we have access to simplex constraints for all $\boldsymbol{x}_k, k \in [n-1]$. As will be seen in the next section, such constraints can be derived by propagating the simplex constraint on the input $\boldsymbol{x}_0$ through the network.

## 3.1 An exact convex relaxation for a single neuron

We begin by considering the simple case of a single neuron, which will form the building block of our final relaxation. Similar to the previous relaxations, we consider the set $\mathcal{S} = \{y, \boldsymbol{x} \mid y = \sigma(\boldsymbol{w}^T \boldsymbol{x} + b), \boldsymbol{x} \in \Delta_m\}$, where $\sigma$ is a convex activation function, for example ReLU. We aim to characterize the convex hull $\mathcal{CH}_\Delta$ of the set $\mathcal{S}$.

**Theorem 3.1.** The convex hull $\mathcal{CH}_\Delta$ of $\mathcal{S}$ is defined by the following convex constraints

$$y \geq \sigma \left( \boldsymbol{w}^T \boldsymbol{x} + b \right), \quad \boldsymbol{x} \in \Delta_m, \tag{4a}$$

$$y \leq \sum_i x_i \left( \sigma \left( \boldsymbol{w}^T \boldsymbol{e}^i + b \right) - \sigma \left( b \right) \right) + \sigma \left( b \right), \tag{4b}$$

where $\boldsymbol{e}^i \in \mathbb{R}^m, e_i^i = 1, e_j^i = 0 \, \forall j \neq i$, denotes the $i$-th coordinate vector in $\mathbb{R}^m$.

By the definition of a convex hull, the proposed relaxation is the tightest possible relaxation for a single neuron with $\boldsymbol{x} \in \Delta_m$. Note that, when $\sigma$ is the ReLU activation function, the only difference between the proposed relaxation and the Planet relaxation $\mathcal{P}_\Delta$ is the upper bound on $y$. Figure 1 compares the upper bound of the Planet relaxation $\mathcal{P}_\Delta$ with the proposed relaxation $\mathcal{CH}_\Delta$ for the case where $m = 2$. As can be seen, our relaxation is significantly tighter than Planet, which paves the way for tractable verification over simplex inputs. The following proposition formally characterizes the difference between the tightness of $\mathcal{CH}_\Delta$ and $\mathcal{P}_\Delta$.

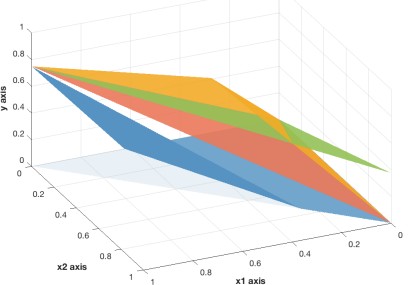

Figure 1: The input domain $\boldsymbol{x} \in \Delta_2$ is shown in light blue, while the output function $y = \text{ReLU}(\boldsymbol{w}^T \boldsymbol{x} + b)$ is shown in blue. It can be seen that the upper bound corresponding to the Planet relaxation (shown in green), is significantly looser in comparison to the upper bound corresponding to the proposed relaxation (shown in orange). Even the Anderson relaxation (shown in yellow) is much looser than our relaxation.

**Proposition 3.2.** For any input dimension $m$, $\mathcal{CH}_\Delta$ is provably tighter than $\mathcal{P}_\Delta$. Specifically, we can characterize the gap between $\mathcal{CH}_\Delta$ and $\mathcal{P}_\Delta$ by the gap in the optimal value of the problem

$$\max_{\boldsymbol{x}} y - y' \text{ subject to } (y, \boldsymbol{x}) \in \mathcal{P}_\Delta, (y', \boldsymbol{x}) \in \mathcal{CH}_\Delta.$$

The gap in the optimal value is proportional to the variance in the weight vector $\boldsymbol{w}$ (see supplementary material for details).

The proofs of the above theorem and proposition are provided in the supplementary material. The proof of Theorem 3.1 utilizes a fundamental result from convex analysis, that any linear function obtains the same maximum value over $\mathcal{S}$ as over its convex hull $\mathcal{CH}_\Delta$.

Note that our relaxation overcomes the convex barrier by providing bounds that are significantly tighter than the Planet relaxation. Convex barrier is a term introduced in Salman et al. [2019], which is defined as the gap between the optimal value of the original verification problem and the optimal convex relaxation of the non-linearity.

**Comparison to Tjandraatmadja et al. [2020]** Our relaxation requires only a linear number of inequalities to describe the convex hull for the composition of a linear function with a convex activation function for simplex inputs. In contrast, the Anderson relaxation [Tjandraatmadja et al., 2020, Anderson et al., 2020] requires an exponential number of constraints. The submodularity based proof for deriving the relaxation from Tjandraatmadja et al. [2020] can help us understand the discrepancy. The method makes use of the Kuhn triangulation of $[0, 1]^n$ [Todd, 1976], which

describes the collection of simplices whose union is $[0, 1]^n$, and requires an exponential number of simplices. Since a unique affine interpolation of the function needs to be constructed on each of these simplices, it requires an overall exponential number of inequalities. In contrast, our relaxation only requires a linear number of inequalities. Note that the input simplex $\Delta_n$ is not one of the simplices from the Kuhn triangulation whose union is the unit hypercube. We provide a visualization for this intuition in the supplementary material.

### 3.2 Final relaxation

Using the convex hull for a single neuron, we can now describe our overall relaxation. Recall that we have assumed $\boldsymbol{x}_k \in \Delta_{n_k}$ for each $k \in [n-1]$. This is ensured through a simplex propagation algorithm described in the next section. In addition, we also compute interval bounds on the pre-activations $\hat{\boldsymbol{x}}_k \in [\ell_k, u_k]$. Using the interval bounds, we define $\theta_k = \frac{u_k}{u_k - \ell_k}$, $\gamma_k = -\ell_k$, $\mathbf{u}_k(\boldsymbol{x}) = \theta_k \odot (\mathbb{L}_k(\boldsymbol{x}) - \gamma_k)$ and $\mathbf{u}'_k(\boldsymbol{x}) = \sum_i \boldsymbol{x}_i (\sigma(\mathbb{L}_k(\boldsymbol{e}^i)) - \sigma(\mathbb{L}_k(\boldsymbol{0}))) + \sigma(\mathbb{L}_k(\boldsymbol{0}))$. Furthermore, since $\hat{\boldsymbol{x}}_n$ is itself a linear function of $\boldsymbol{x}_{n-1}$, with a slight overload of notation, we denote the objective function of our relaxation as $\Psi(\boldsymbol{x}_{n-1})$. Using the above notation, the overall proposed convex relaxation of problem (1) for the ReLU activation function can be written as

$$\min_{\boldsymbol{x}} \quad \Psi(\boldsymbol{x}_{n-1}) \tag{5a}$$

$$s.t. \quad \boldsymbol{x}_k \geq \mathbb{L}_k(\boldsymbol{x}_{k-1}), \boldsymbol{x}_k \geq 0 \qquad k \in [n-1], \text{ (Planet lower bound)} \tag{5b}$$

$$\boldsymbol{x}_k \leq \mathbf{u}_k(\boldsymbol{x}_{k-1}), \qquad k \in [n-1], \text{ (Planet upper bound)} \tag{5c}$$

$$\boldsymbol{x}_k \leq \mathbf{u}'_k(\boldsymbol{x}_{k-1}), \qquad k \in [n-1], \quad \text{(from equation 4b)} \tag{5d}$$

$$\boldsymbol{x}_k \in \Delta \qquad k \in \{0\} \cup [n-1]. \tag{5e}$$

Similar relaxations can be derived for other convex activation functions by linearizing the convex function around a given point. However, we focus on the ReLU case for brevity of exposition, particularly since ReLU is the most commonly used convex activation function.

## 4 Algorithm

Our verification algorithm is composed of two phases: (i) propagating simplex constraints on the activations at every layer; and (ii) computing a lower bound on problem (1). We describe these in the following subsections.

### 4.1 Simplex propagation

We assume that the output of each layer is non-negative. If this is not the case, we can simply add a constant to the activation of each layer so that the output becomes non-negative. Let us denote $h_k(\boldsymbol{x}) = \sigma(\mathbb{L}_k(\boldsymbol{x}))$. Since $h_k(.)$ is an element-wise convex function of its inputs, $\mathbf{1}^T h_k(.)$ is also a convex function. Using the fact that the maximum of a convex function over the simplex is attained at one of the vertices, it can be shown that the following inequality holds true

$$\sum_i x_{ki} \leq \max_{j \in \{0, \dots, n_{k-1}\}} \mathbf{1}^T h_k(\boldsymbol{e}^j) = \alpha_k. \tag{6}$$

Here, $x_{ki}$ denotes the $i$-th coordinate of the vector of activations $x_k$ at the output of layer $h_k$.

Note that the above inequality can be rewritten in the form $\sum_i \tilde{x}_{ki} \leq 1$, where $\tilde{x}_{ki} = \frac{x_{ki}}{\alpha_k}$, so that we can propagate simplex-like constraint simply by rescaling activations at the layer appropriately. Details for conditioning the intermediate layers into simplex using inequalities of the above form are provided in the supplementary material.

### 4.2 Efficient solver

The optimization problem (5) is a Linear Program and as such can be solved easily by off-the-shelf solvers [Gurobi]. However, given the size of modern deep networks, such an approach would struggle to scale. Hence, researchers have recently started developing scalable custom solvers for relaxations of neural network outputs.

Zhang et al. [2018], Singh et al. [2019] developed efficient solvers for relaxations that rely on bounds on the output of a neuron that are expressed as linear functions of the input. Such solvers have been scaled to be extremely efficient and amenable to use within branch and bound frameworks [Wang et al., 2021]. Inspired by their success, we extend their capabilities to solve our novel tight relaxations derived in the previous section. To this end, we first relax problem (5) to a form that can be solved efficiently using a single backward pass. Concretely, we combine the lower bounds (5b) and the non-negativity constraint from (5e) into a single lower bound using weighting coefficients $\underline{a}_k$. We also combine the upper bounds (5c and 5d) into a single upper bound with weighting coefficients $\overline{a}_k$. The values of the weighting coefficients $\underline{a}_k$ and $\overline{a}_k$ are constrained to lie between 0 and 1. This leads to the following optimization problem

$$L(\underline{a}, \overline{a}) = \min_{\boldsymbol{x}} \quad \Psi\left(\boldsymbol{x}_{n-1}\right) \tag{7a}$$

$$s.t. \quad \boldsymbol{x}_0 \in \Delta \ , \tag{7b}$$

$$\boldsymbol{x}_k \geq \underline{a}_k \odot \mathbb{L}_k\left(\boldsymbol{x}_{k-1}\right) \qquad\qquad k \in [n-1], \tag{7c}$$

$$\boldsymbol{x}_k \leq \overline{a}_k \odot \mathbf{u}_k\left(\boldsymbol{x}_{k-1}\right) + \left(1 - \overline{a}_k\right) \odot \mathbf{u}'_k\left(\boldsymbol{x}_{k-1}\right) \qquad k \in [n-1]. \tag{7d}$$

It turns out that the above formulation can be solved efficiently by a single backward pass over the network. In order to derive the solution, it is useful to introduce the notion of decomposition of an affine function into monotone, anti-monotone and constant parts.

**Definition 4.1.** For any scalar-valued affine function $f\left(\boldsymbol{x}\right)$, define

$$f^+\left(\boldsymbol{x}\right) = \left(\max\left(\frac{\partial f}{\partial \boldsymbol{x}}, 0\right)\right)^T \boldsymbol{x}, f^-\left(\boldsymbol{x}\right) = \left(\min\left(\frac{\partial f}{\partial \boldsymbol{x}}, 0\right)\right)^T \boldsymbol{x}, f^c = f\left(0\right), \tag{8}$$

so that $f\left(\boldsymbol{x}\right) = f^+\left(\boldsymbol{x}\right) + f^-\left(\boldsymbol{x}\right) + f^c$.

Since $f^+$ only involves the positive coefficients in the gradient of $f$, it is monotonically increasing, and similarly, $f^-$ is monotonically decreasing. Exploiting this and the fact that $\Psi$ is a scalar valued linear function, we obtain

$$\Psi\left(\boldsymbol{x}_{n-1}\right) = \Psi^+\left(\boldsymbol{x}_{n-1}\right) + \Psi^-\left(\boldsymbol{x}_{n-1}\right) + \Psi_0$$
$$\geq \Psi^-\left(\overline{a}_{n-1} \odot \mathbf{u}_{n-1}\left(\boldsymbol{x}_{n-2}\right) + \left(1 - \overline{a}_{n-1}\right) \odot \mathbf{u}'_{n-1}\left(\boldsymbol{x}_{n-1}\right)\right) + \Psi^+\left(\underline{a}_{n-1} \odot \mathbb{L}_{n-1}\left(\boldsymbol{x}_{n-2}\right)\right)$$
$$+ \Psi^c. \tag{9}$$

The lower bound above is an affine function of $\boldsymbol{x}_{n-2}$ and hence leads to a recursive algorithm where we compute lower bounds on the specification expressed as a function of $\boldsymbol{x}_{k-1}$ given a lower bound on the specification expressed as a linear function of $\boldsymbol{x}_k$.

Therefore, this leads to a recursive algorithm for computing a lower bound on the optimal value of (7), that is presented in the Subroutine SIMPLEX_BACKWARD function on line 11 in Algorithm 1. It can in fact be proven that this is the exact optimal value, using an argument similar to Salman et al. [2019], Section 3.

The computation required to express a lower bound on the specification given as a linear function of $\boldsymbol{x}_{k+1}$ to a lower bound expressed as a linear function of $\boldsymbol{x}_k$ is shown on line (14) of Algorithm 1. Note that this computation can be conveniently done in frameworks that support automatic differentiation, by computing the gradient of the expression on the right hand side of line 14 at 0 and its value at 0, which gives the vector of coeffients and the bias term of the affine function of $\boldsymbol{x}_k$. Thus, the cost of implementing line 14 of the algorithm is equal to the cost of performing backpropagation through a single layer of the network.

Denote the projection onto the unit hypercube as $\pi\left(\boldsymbol{x}\right) = \min\left(\max\left(\boldsymbol{x}, 0\right), 1\right)$. Our overall algorithm is presented in Algorithm 1. It consists of two main functions SIMPLEX_BACKWARD and SIMPLEX_VERIFY. The SIMPLEX_BACKWARD algorithm computes $L\left(\underline{a}, \overline{a}\right)$ for a fixed value of $\underline{a}, \overline{a}$. However, this lower bound is valid for any choice of these parameters. Hence SIMPLEX_VERIFY performs projected gradient ascent on $L$ with respect to $\underline{a}, \overline{a}$ to obtain the tightest possible lower bound on the (equation 1). The idea of optimizing the weighting coefficients $\underline{a}, \overline{a}$ to obtain tighter bounds is inspired from the algorithm of Xu et al. [2021], with suitable adaptations to the setting involving our novel relaxation.

**Proposition 4.2.** Algorithm 1 computes a lower bound on the optimal value of (1).

*Proof.* Follows by recursively applying the lower bound at each layer while defining $f_k$. □

**Computational Complexity:** The complexity of SIMPLEX_BACKWARD function 11 for computing bounds for a given value of $\underline{a}^t, \bar{a}^t$, is the same as the cost of two backward passes through the network. This is twice the cost of CROWN [Zhang et al., 2018], or one iteration of auto-lirpa [Xu et al., 2021], which uses a single backward pass.

---

**Algorithm 1** Simplex Verify

---
 1: **function** SIMPLEX_VERIFY($\Psi$)
 2:     Initialise $\underline{a}$ and $\bar{a}$ with values between $\mathbf{0}$ and $\mathbf{1}$
 3:     **for** $t \in [\![0, t_{max} - 1]\!]$ **do**
 4:         $L(\underline{a}^t, \bar{a}^t) = $ SIMPLEX_BACKWARD$(\Psi, \underline{a}^t, \bar{a}^t)$
 5:         Compute gradients $\frac{dL}{d\underline{a}^t}, \frac{dL}{d\bar{a}^t}$ via backpropagation
 6:         $\underline{a}^{t+1}, \bar{a}^{t+1} \leftarrow$ update gradient ascent (or Adam)
 7:         $\underline{a}^{t+1}, \bar{a}^{t+1} \leftarrow \pi\left(\underline{a}^{t+1}\right), \pi(\bar{a}^{t+1})$        (projection)
 8:     **end for**
 9:     **return** $L(\underline{a}^{t+1}, \bar{a}^{t+1})$
10: **end function**
11: **function** SIMPLEX_BACKWARD($\Psi, \underline{a}, \bar{a}$)
12:     $f_N \leftarrow \Psi$
13:     **for** $k \in [\![n - 1, 0]\!]$ **do**
14:         Set $f_k(\boldsymbol{x}) \leftarrow f_{k+1}^-(\bar{a}_k \odot \mathbf{u}_k(\boldsymbol{x}) + (1 - \bar{a}_k) \odot \mathbf{u}_k'(\boldsymbol{x})) + f_{k+1}^+(\underline{a}_k \odot \mathbb{L}_k(\boldsymbol{x})) + f_{k+1}^c$.
15:     **end for**
16:     $L(\underline{a}, \bar{a}) = \min_{\boldsymbol{x}_0 \in \Delta} f_0(\boldsymbol{x}_0)$
17:     **return** $L(\underline{a}, \bar{a})$
18: **end function**

---

## 5 Experiments

In this section, we demonstrate the effectiveness of the proposed method on two specifications: (i) robustness to $\ell_1$ perturbations for image classifiers (Sec 5.1), and (ii) robustness of multi-modal classifiers to text perturbations (Sec 5.2).

### 5.1 $\ell_1$ robustness verification

**Experimental Setup** We verify the $\ell_1$ robustness of networks from [de Palma et al., 2021a, Bunel et al., 2020b] and VNN-COMP [2020]. We compute the lower bound on the robustness margin (difference between the ground truth logit and the other logits) using the verification methods. An image is said to be verified if the lower bound across all possible labels is positive. We evaluate the effectiveness of various methods for incomplete verification on the MNIST [Lecun] and CIFAR-10 [Krizhevsky and Hinton, 2009] datasets. For MNIST, we evaluate on the entire test set, and for CIFAR-10 we evaluate on 1000 random images from the test set [Krizhevsky and Hinton, 2009]. The MNIST and CIFAR-10 datasets are widely used in the machine learning community, and are available under the creator's consent and MIT license respectively. For both MNIST and CIFAR-10, we use the model architectures from [de Palma et al., 2021a]. The models are trained using the SLIDE attack (sparse $\ell_1$-descent attack) from Tramer and Boneh [2019] with $\epsilon = 0.3$ for all networks except the VNN-comp big network, which is trained with $\epsilon = 0.05$. We used the publicly available training implementation of [Ding et al., 2019] (see supplementary material for details). The code is made available under the LGPL License online [1]. We also test on the SGD trained CIFAR Wide model from [de Palma et al., 2021a]. We verify robustness against input perturbations lying in $\ell_1$ norm ball with $\epsilon = 0.35$ for the MNIST network, $\epsilon = 0.2$ for the VNN-comp big network and $\epsilon = 0.5$ for all the other CIFAR-10 networks.

---
[1] https://github.com/BorealisAI/advertorch.

| Dataset | MNIST | CIFAR-10 | | | | | | |
|---|---|---|---|---|---|---|---|---|
| Model | | OVAL Wide | OVAL Base | OVAL Wide | OVAL Deep | OVAL Wide | VNN Med | VNN Big |
| Training | | SLIDE | SLIDE | SLIDE | SLIDE | SGD | SLIDE | SLIDE |
| **Accuracy** Nominal | | 98.8% | 75.1% | 79.3% | 72.1% | 74.4% | 81.4% | 83.6% |
| Pgd | | 98.2% | 73.5% | 77.0% | 69.8% | 73.3% | 80.5% | 82.3% |
| **Verified Accuracy** Gurobi Planet | | 31.7% | 34.1% | 18.4% | 11.1% | 13.5% | - | - |
| Gurobi Simplex | | 45.2% | 48.6% | 29.4% | 13.4% | 23.7% | - | - |
| Opt-Lirpa Planet | | 31.0% | 33.6% | 17.9% | 10.8% | 13.5% | 48.8% | 60.0% |
| Simplex Verify | | 44.6% | 48.0% | 28.8% | 13.4% | 22.4% | 59.4% | 66.4% |
| **Verified Time/Sample** Gurobi Planet | | 74.61s | 22.80s | 114.92s | 86.84s | 114.70s | - | - |
| Gurobi Simplex | | 72.47s | 22.95s | 72.17s | 59.22s | 70.42s | - | - |
| Opt-Lirpa Planet | | 0.04s | 0.04s | 0.04s | 0.06s | 0.04s | 0.05s | 0.06s |
| Simplex Verify | | 0.04s | 0.04s | 0.04s | 0.05s | 0.04s | 0.05s | 0.06s |

Table 1: Verified accuracy and verification time of different solvers on MNIST and CIFAR-10 models. We test on the entire test set for MNIST, and random 1000 test images for CIFAR-10. Simplex Verify denotes our proposed solver. Our proposed method achieves much higher verified accuracy in comparison to the state of the art baseline, in the same amount of time. '-' denotes instances not solved within a 5 minute timeout.

**Methods**    We compare against other propagation based solvers and Gurobi. Gurobi baselines employ the commercial black-box solver Gurobi [Gurobi Optimization, 2020]. Gurobi solves the problems to optimality, giving the tightest possible bounds for the corresponding relaxations. Gurobi Planet corresponds to solving the Planet relaxation [Ehlers, 2017] of the network. Gurobi Simplex corresponds to solving our relaxation using Gurobi. Both the methods are run on 4 CPU threads on an Intel(R) Core(TM) i7-4960X CPU @ 3.60GHz processor. We also compare against an optimized LiRPA solver for the Planet relaxation [Ehlers, 2017], and refer to it as Opt-Lirpa Planet. The solver remains the same as our solver, with the only difference being that the upper bound corresponding to our relaxation is not present (see supplementary material for details). Note that the intermediate layer bounds in Opt-Lirpa Planet are not jointly optimized. Simplex Verify corresponds to our solver described in Sec 4.2. Both the LiRPA based solvers use Adam [Kingma and Ba, 2015] for updating the weighting vectors $a$, and are run on a single Nvidia Titan Xp GPU. All the methods use the same intermediate bounds, which are computed using Opt-Lirpa Planet run for 20 iterations. We compare the effectiveness of the different methods for computing the final layer bounds. Further details about the baselines and experimental settings, including the hyper-parameters, are provided in the supplementary material.

**Results**    Table 1 shows the verified accuracy and average verification time per sample of different methods. Gurobi Simplex achieves much higher verified accuracy (up to $16.5\%$ higher) in comparison to Gurobi Planet. This is in line with Theorem 3.1 and Proposition 3.2 because our proposed relaxation is much tighter than Planet. This also shows the benefit of using tighter relaxations. For a fair comparison, the number of iterations of both LiRPA based methods were tuned such that each of them take the same amount of time. We tuned the number of iterations on a subset of images. It is worth noting that the proposed solver, Simplex Verify, achieves much higher verified accuracy than Opt-Lirpa Planet, in the same amount of time. Simplex Verify also achieves comparable accuracy to Gurobi Simplex, while being nearly 3 orders of magnitudes faster, which shows the effectiveness of the proposed solver in solving Problem 5.

We also compare the tightness of the bounds achieved by different methods. We obtained the lower bounds provided by different solvers. We can get the upper bounds using the sparse PGD attack, SLIDE [Tramer and Boneh, 2019]. A smaller gap between the PGD upper bound and verified lower bound, indicates tighter verification. Figure 2a shows the pointwise comparison of the gap to PGD for Gurobi Planet and Gurobi Simplex, on the same data. PGD gap is much smaller for Gurobi Simplex in comparison to Gurobi Planet, thereby showing that our relaxation achieves much tighter verification. Figure 2b shows the pointwise comparison of the gap for Opt-Lirpa Planet and our Simplex Verify. Simplex Verify achieves much tighter verification.

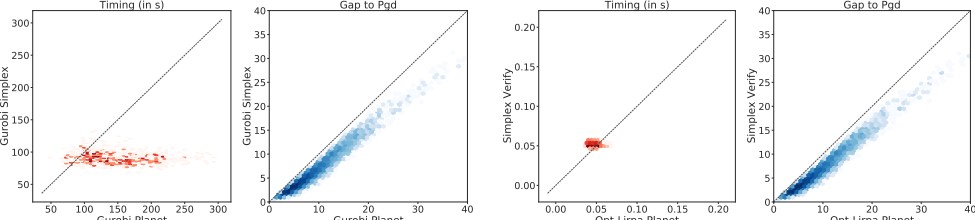

(a) Comparison of runtime (left) and gap of lower bounds to PGD upper bounds (right), for Gurobi Simplex and Gurobi Planet.

(b) Comparison of runtime (left) and gap of lower bounds to PGD upper bounds (right), for Simplex Verify and Opt-Lirpa Planet.

Figure 2: Pointwise comparison of the lower bounds on CIFAR-10 test set on the adversarially trained Wide model, for a subset of the methods. Darker colour shades mean higher point density (on a logarithmic scale). The oblique dotted line corresponds to the equality.

## 5.2 Multi-modal classifier robustness verification

**Experimental Setup** In this section, we are interested in verifying the robustness of multi-modal classifiers to text-based attacks. The motivation are scenarios where image based models are augmented with text to improve the accuracy of the classifier. However, this makes the model vulnerable to text based attacks. The aim of this experiment is to show the adversarial vulnerability of such models and verify their robustness. For this experiment we verify the robustness of models on the UPMC FOOD-101 dataset [Wang et al., 2015], which is a commonly used dataset for multi-modal classification (see [Kiela et al., 2019]). This dataset consists of images and recipes of different food items. It is made available by the creators' consent online [2]. The specification that we are interested in verifying is as follows: for a given image and text pair, only the text is perturbed and any possible text from the given vocabulary is allowed in the attack. The aim of this specification is to characterise the worst-case sensitivity of the model. We are not aiming for perfect robustness to the noise in text, but aim to check its sensitivity. We use a ConcatBOW model for this task as proposed in Kiela et al. [2019]. The model extracts an image embedding using a standard pretrained ResNet-152 model. It also extract a Bag of words embedding for the text. The model concatenates the embeddings and feeds them into a multilayer perceptron (MLP) classifier, which has 1,846,200 parameters. Arbitrary changes in the text can be modelled as a simplex as described in Section 2.1. The model follows the same architecture as the state-of-the-art model for this dataset [Ignazio Gallo and Grassa, 2020], except that we replace Bert embeddings with BOW embeddings. We reduce the dataset from 101 classes to 10 classes. More details are provided in the supplementary.

**Related Work** Huang et al. [2019] considered verification against synonym replacements or character flip perturbations on text classification models. The input specification was modeled as a simplex and they proposed to use Interval bound propagation (IBP) for the same. Concurrent to Huang et al. [2019], Jia et al. [2019] considered a specification where every word in the text can be replaced with a similar word and used IBP. Xu et al. [2020] use more recent LiRPA variants for verification of synonym-based word substitution with a maximum of 6 word substitutions. Concurrent to our work, Bonaert et al. [2021] proposed a new method for certifying Transformers to synonym based attacks. They provide a lifting from $\ell_\infty$ analysis techniques to other norms, and propose a new convex relaxation for a number of settings, but none of these characterize the convex hull exactly in any setting. In contrast to these works, we consider a much more challenging specification where any possible text from the vocabulary is allowed. This specification includes arbitrary length sentences. Further, even our baseline, Opt-Lirpa Planet, is much tighter than IBP. We also propose a tighter relaxation which characterises the convex hull for our setting of a composition over a convex activation function and an activation function where the input is constrained to be in the simplex, and propose an efficient algorithm for the relaxation. It is also important to note that we perform experiments on verification-agnostic networks unlike Huang et al. [2019].

**Results** We compare the lower bounds on the robustness margin for Opt-Lirpa Planet and Simplex Verify. We use PGD attack for computing the upper bound. The networks are trained to be robust to simplex perturbations using PGD training. Figure 3 shows the pointwise comparison of the gap to

---

[2]http://visiir.lip6.fr/

PGD upper bounds for Opt-Lirpa Planet and Simplex Verify. Note here that a smaller gap indicates tighter verification. For a fair comparison, the iterations of the propagation algorithms were again tuned such that each of them takes the same amount of time (0.08s). It can be seen that Simplex Verify achieves much tighter bounds in the same amount of time.

In Table 2, we present the verified accuracy achieved by the different methods with the nominal and Pgd accuracy. We present results for networks trained with different weighting for nominal and PGD loss during the adversarial training. Network trained only on images achieves $87.82\%$ accuracy. Note that our method achieves remarkably higher verified accuracy, up to $25\%$ higher, as compared to Opt-Lirpa Planet in the same amount of time. This shows the benefit of our approach over the existing solvers.

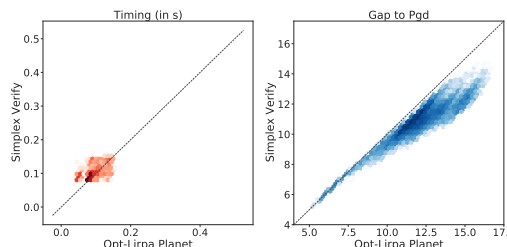

Figure 3: Pointwise comparison of runtime (left) and gap of lower bounds to PGD upper bounds (right) for the global specification on Food-101 dataset.

# 6 Discussion and Broader Impact

As machine learning continuously finds new application domains, the robustness of machine learning systems in the face of adversarsarial behavior becomes increasingly relevant. Neural network verification, which seeks to obtain provable guarantees on robustness of neural networks, has mostly focused on $\ell_\infty$ perturbations. However, real security threats often involve more drastic changes that could lead to arbitrary perturbations in the input. In this paper, we have we have proposed an efficient algorithm for verification of neural networks against simplex constrained perturbations. We have shown

|  |  |  |  |  |
|---|---|---|---|---|
| Accuracy | Nominal | 86.3% | 87.3% | 92.3% |
|  | Pgd | 84.7% | 84.5% | 19.2% |
| Verified | Opt-Lirpa Planet | 17.9% | 08.0% | 01.2% |
| Accuracy | Simplex Verify | 42.2% | 32.8% | 01.3% |

Table 2: Verified accuracy achieved by different solvers on the 2164 test images of the reduced Food-101 dataset. Results are presented for networks trained with different weighting for nominal and PGD loss. Our method (Simplex Verify) achieves up to $24\%$ higher verified accuracy, in the same amount of time as the state of the art baseline Opt-Lipra Planet.

the practical importance of this specification through two applications. Firstly, we verify the robustness of image classifiers to $\ell_1$ perturbations, which could be an appropriate threat model when perturbations of different features are correlated. Secondly, we consider a challenging specification, whereby any arbitrary text perturbations are allowed on multi-modal (image and text input) classifiers. This highlights the vulnerability of multi-modal classifiers to text attacks. The proposed algorithm improves the state of the art verification accuracy by up to $25\%$.

Multi-modal data is used in various domains including social media, and the internet in general. We believe that our work could find application in verifying classifiers which label advertisements and social media posts as illegal or hateful. Although we tackle arbitrary perturbation in the text input, our work does not verify against all possible perturbations on the multi-modal input. Another limitation is that we have considered only a particular class of multi-modal architectures. It is also important to verify other commonly used architectures. Further, verification exposes flaws in neural network models, which on the one hand can help improve their robustness, but on the other hand, can be exploited by an adversary.

**Acknowledgements and Disclosure of Funding**

Harkirat was supported using a Tencent studentship through the University of Oxford. Philip H.S. Torr was supported by the EPSRC grant: Turing AI Fellowship: EP/W002981/1, EPSRC/MURI grant EP/N019474/1 and the Royal Academy of Engineering. We also thank Rudy Bunel for feedback on the draft.

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
