# Overcoming the Convex Barrier for Simplex Inputs: Supplementary Material

**Harkirat Singh**
University of Oxford
harkirat@robots.ox.ac.uk

**M. Pawan Kumar**
DeepMind
mpawan@deepmind.com

**Philip H.S. Torr**
University of Oxford
phst@robots.ox.ac.uk

**Krishnamurthy (Dj) Dvijotham**
DeepMind
dvij@google.com

# Appendix

## 1 Technical Appendix

### 1.1 Proofs

In this section, we present proofs for the theoretical results presented in the main paper Section 3.1.

**Theorem 1.1.** The convex hull of set $\mathcal{S}$, $\mathcal{CH}(\mathcal{S})$, is the set of all $(y, \boldsymbol{x})$ satisfying the following convex constraints:

$$y \geq \sigma\left(\boldsymbol{w}^T\boldsymbol{x} + b\right), \quad \boldsymbol{x} \in \Delta, \tag{1a}$$

$$y \leq \sum_i x_i \left(\sigma\left(\boldsymbol{w}^T\boldsymbol{e}^i + b\right) - \sigma\left(b\right)\right) + \sigma\left(b\right), \tag{1b}$$

where $\boldsymbol{e}^i \in \mathbb{R}^n, e_i^i = 1, e_j^i = 0 \,\forall j \neq i$, denotes the $i$-th coordinate vector in $\mathbb{R}^n$.

*Proof.* In this proof, for brevity, we write $\mathcal{CH}$ to denote $\mathcal{CH}\left(\mathcal{S}\right)$. To begin with, we note that $\mathcal{CH}$ is clearly a convex set, since $\sigma$ is a convex function.

The convex hull of the set

$$\mathcal{S} = \{(\boldsymbol{x}, y) : y = \sigma\left(\boldsymbol{w}^T\boldsymbol{x} + b\right), \boldsymbol{x} \in \Delta\}$$

can be characterized as the intersection of all the halfspaces that contain it [Boyd and Vandenberghe, 2004]. Equivalently, if we demonstrate that any linear function obtains the same maximum value over $\mathcal{S}$ as over $\mathcal{CH}$, the proof is complete.

Let $\alpha \in \mathbb{R}^n, \beta \in \mathbb{R}$ denote coefficients of a linear function on $\mathcal{S}$.

If $\beta > 0$, the maximum is given by

$$\max_{(\boldsymbol{x},y)\in\mathcal{S}} \alpha^T\boldsymbol{x} + \beta y = \max_{\boldsymbol{x}\in\Delta} \alpha^T\boldsymbol{x} + \beta\sigma\left(\boldsymbol{w}^T\boldsymbol{x} + b\right) = \max_{e\in\mathcal{V}} \alpha^T e + \beta\sigma\left(\boldsymbol{w}^T e + b\right)$$

where $\mathcal{V} = \{0, \boldsymbol{e}^1, \boldsymbol{e}^2, \dots, \boldsymbol{e}^n\}$ denotes the set of vertices of the simplex $\Delta$, and the second inequality follows from the fact that the objective is a convex function of $\boldsymbol{x}$. Since the constraint is invariant to scaling $\beta$, we can assume $\beta = 1$. Thus, we have that for any $(\boldsymbol{x}, y) \in \mathcal{S}$,

$$\max_{(\boldsymbol{x},y)\in\mathcal{S}} \alpha^T\boldsymbol{x} + y = \sigma\left(b\right) + \max\left(\max_i \alpha_i + \sigma\left(\boldsymbol{w}^T\boldsymbol{e}^i + b\right) - \sigma\left(b\right), 0\right)$$

35th Conference on Neural Information Processing Systems (NeurIPS 2021).

If we optimize the same linear function over $\mathcal{CH}$, we obtain

$$\max_{(\boldsymbol{x},y)\in\mathcal{CH}}\alpha^T\boldsymbol{x}+y=\max_{\boldsymbol{x}\in\Delta}\alpha^T\boldsymbol{x}+\sum_i x_i\left(\sigma\left(\boldsymbol{w}^T\boldsymbol{e}^i+b\right)-\sigma\left(b\right)\right)+\sigma\left(b\right)$$

$$=\max_{\boldsymbol{x}\in\Delta}\sum_i x_i\left(\alpha_i+\sigma\left(\boldsymbol{w}^T\boldsymbol{e}^i+b\right)-\sigma\left(b\right)\right)+\sigma\left(b\right)$$

$$=\sigma\left(b\right)+\max\left(\max_i\left(\alpha_i+\sigma\left(\boldsymbol{w}^T\boldsymbol{e}^i+b\right)-\sigma\left(b\right)\right),0\right)$$

Thus the linear functions obtain the same optimum over both sets when $\beta>0$.

If $\beta<0$, the maximum value of the linear function over $\mathcal{S}$

$$\max_{(\boldsymbol{x},y)\in\mathcal{S}}\alpha^T\boldsymbol{x}+\beta y=\max_{\boldsymbol{x}\in\Delta}\alpha^T\boldsymbol{x}+\beta\sigma\left(\boldsymbol{w}^T\boldsymbol{x}+b\right)=\max_{(\boldsymbol{x},y)\in\mathcal{CH}}\alpha^T\boldsymbol{x}+\beta y$$

where the last equality follows from the fact that $\beta<0$ and $\mathcal{CH}$ only contains one lower bound on $y$ for any fixed $\boldsymbol{x}$, ie, $y\geq\sigma\left(\boldsymbol{w}^T\boldsymbol{x}+b\right)$ (equation 1b) and the objective is optimized by setting $y$ to its lower bound.

The only remaining case is when $\beta=0$ and in this case, the optimization over both $\mathcal{S},\mathcal{CH}$ reduce to

$$\max_{\boldsymbol{x}\in\Delta}\alpha^T\boldsymbol{x}=\max_{e\in\mathcal{V}}\alpha^T e$$

and are indeed identical.

Thus, we have demonstrated that the maximum value of any linear function over $\mathcal{S}$ and over $\mathcal{CH}$ are identical, and hence $\mathcal{CH}$ is the convex hull of $\mathcal{S}$. □

**Proposition 1.2.** For any input dimension $m$, $\mathcal{CH}_\Delta$ is provably tighter than $\mathcal{P}_\Delta$. Specifically, we can characterize the gap between $\mathcal{CH}_\Delta$ and $\mathcal{P}_\Delta$ by the gap in the optimal value of the problem

$$\max y-y'\text{ subject to }(y,\boldsymbol{x})\in\mathcal{P}_\Delta,(y',\boldsymbol{x})\in\mathcal{CH}_\Delta. \tag{2}$$

The gap in the optimal value is shown to be proportional to the variance in the weight vector $\boldsymbol{w}$.

*Proof.* Let $\boldsymbol{x}\in\Delta_n$, $h(\boldsymbol{x})=\text{ReLU}(\boldsymbol{w}^T\boldsymbol{x}+b)$. Let $w_{\min}=\min_i w_i$. Note here that $w_{\min}$ includes comparison to 0 to take care of the origin point of the simplex. Let $i_{\min}$ or $i_{\max}$ denote the indices corresponding to $w_{\min}$ and $w_{\max}$ respectively. We use the pre-activation bounds computed using simplex, as per the definition of $\mathcal{P}_\Delta$, which is $l=w_{\min}+b$ and $u=w_{\max}+b$. We assume that $l\leq 0\leq u$. The upper bounding cut in planet relaxation is

$$y^{\mathcal{P}}(\boldsymbol{x})=\frac{u}{u-l}(\boldsymbol{w}^T\boldsymbol{x}+b-l).$$

We can write this as

$$y^{\mathcal{P}}(\boldsymbol{x})=\frac{w_{\max}+b}{w_{\max}-w_{\min}}(\boldsymbol{w}^T\boldsymbol{x}-w_{\min}).$$

To compare the tightness, we compare the difference $y^{\mathcal{P}}-h(\boldsymbol{x})$ for the simplex vertices. We begin by noting that the value $y^{\mathcal{P}}$ at a vertex point $\boldsymbol{e}^i$ will be

$$y^{\mathcal{P}}(\boldsymbol{e}^i)=\frac{w_{\max}+b}{w_{\max}-w_{\min}}(w_i-w_{\min}). \tag{3}$$

By the definition and construction of the Planet relaxation, $y^{\mathcal{P}}=h(.)$ when $i=i_{\min}$ or $i=i_{\max}$. For $i\in\{0,..,n\}$ we have

$$y^{\mathcal{P}}-h\left(\boldsymbol{e}^i\right)=\begin{cases}-l\dfrac{w_{\max}-w_i}{w_{\max}-w_{\min}} & \text{if }w_i+b\geq 0, \tag{4a}\\[2ex] u\dfrac{w_i-w_{\min}}{w_{\max}-w_{\min}} & \text{if }w_i+b<0. \tag{4b}\end{cases}$$

Since the difference $y^{\mathcal{P}}-h\left(\boldsymbol{e}^i\right)$ will be non-zero, we can conclude that the planet relaxation does not represent the convex hull. Since our relaxation represents the convex hull (see Theorem 1.1), thus it follows that our relaxation is tighter than the Planet relaxation.

Let $y^{\mathcal{CH}}$ denote the upper bound corresponding to our proposed relaxation. Note that at the simplex vertices $e^i$, $y^{\mathcal{CH}} = h(e^i)$. Thus at these vertices, the gap $y^{\mathcal{P}} - y^{\mathcal{CH}}$ can be given by Equation 4. This difference provides a valid lower bound to the gap in Equation 2. Note that this gap is proportional to the variance in the weight vector $\boldsymbol{w}$. $\qquad\square$

## 1.2 Comparison to Anderson relaxation [Anderson et al., 2020]

To compare the tightness with respect to the Anderson relaxation [Anderson et al., 2020, Tjandraat-madja et al., 2020], we use a two dimensional example. The example corresponds to the weights in Figure 1 of the main paper. We use $\boldsymbol{x} \in \Delta_2$ with $w = [2, 1]$, $b = -1.25$ and $y = \text{ReLU}(w^T\boldsymbol{x} + b)$. To derive the anderson relaxation for this two dimensional setting, we take inspiration from Example 1 in Appendix of Tjandraatmadja et al. [2020]. Figure 1 in our main paper compares the tightness between different relaxations. It can be seen that the Anderson relaxation does not describe the convex hull, and is much looser than our proposed relaxation.

For using the Anderson relaxation, we need to replace the input simplex with the unit hypercube. The relaxation first uses Kuhn triangulation of $[0, 1]^n$ [Todd, 1976], which is used to describe the collection of simplices whose union is $[0, 1]^n$. It requires an exponential number of simplices to describe the unit hypercube. The method then constructs a unique affine interpolation of the function $h(x)$ on each of these simplices. This gives an overall exponential number of inequalities. For our two dimensional example, we need to divide the unit square into two triangles $\mathbf{T}_1$ and $\mathbf{T}_2$ as shown in Figure 1. Then constraints $r_1$ and $r_2$ are constructed using these two triangles. In contrast, our relaxation works directly with the input simplex and only requires one upper constraint. Figure 1 shows a visualization for this intuition behind why our relaxation only requires a linear number of constraints in comparison to the exponential constraints of the Anderson relaxation [Anderson et al., 2020]. To show a direct correspondence, we construct this figure by modifying Figure 3 from Appendix of Tjandraatmadja et al. [2020].

## 1.3 Implementation Details

In this section, we provide implementation details for various components of the method.

**Conditioning from $\ell_1$ to simplex** Here we show how $\ell_1$ norm perturbations on images can be modelled as simplex perturbations. Let $\boldsymbol{x} \in \mathbb{R}^m$ denote the image input space and let the input perturbations lie within an $\ell_1$ ball: $\{\boldsymbol{x} \mid \left\|\boldsymbol{x} - \boldsymbol{x}^0\right\|_1 \le \epsilon\}$. This input domain can be reformulated as a simplex as

$$\boldsymbol{x} = \boldsymbol{x}^0 + \epsilon M \boldsymbol{z}, \ \boldsymbol{z} \in \Delta_{2m}, \ \underset{(m \times 2m)}{M} = \begin{pmatrix} 1 & -1 & 0 & 0 & \dots & 0 & 0 \\ 0 & 0 & 1 & -1 & \dots & 0 & 0 \\ \vdots & & & & & & \\ 0 & 0 & 0 & 0 & \dots & 1 & -1 \end{pmatrix}. \tag{5}$$

This transformation can be achieved by conditioning the first layer as

$$W\boldsymbol{x} + b = W(\boldsymbol{x}^0 + \epsilon M \boldsymbol{z}) + b \tag{6a}$$
$$= \epsilon W M \boldsymbol{z} + (W\boldsymbol{x}^0 + b) \tag{6b}$$
$$= W'\boldsymbol{z} + b', \tag{6c}$$

where $W'$ and $b'$ denote the weights and bias of the conditioned layer whose input lies in a simplex.

**Conditioning intermediate layers** In Section 4.1 of the main paper, we proposed a technique to propagate simplex constraints throughout the network. We derived inequalities of the following form on the intermediate layers:

$$\sum_i x_{k,i} \le \max_{j \in \{0,\dots,n_{k-1}\}} \mathbf{1}^T h_k\left(e^j\right) = \alpha_k. \tag{7}$$

Here, $x_{k,i}$ denotes the $i$-th coordinate of the vector of activations $x_k$ at the output of layer $h_k$. Here we show how to condition the activations of the layers to propagate simplex using the above inequality, assuming that the non-linearity $\sigma$ is ReLU.

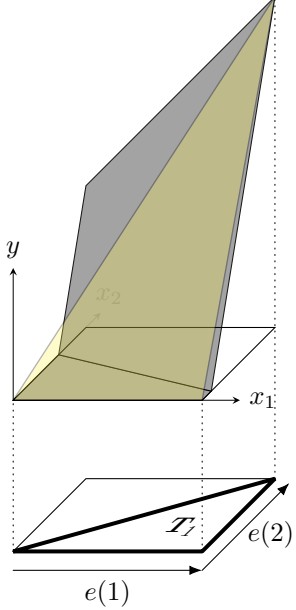

(a) Constructing $r_1$ (yellow) using $\mathbf{T}_1$ from the Anderson relaxation. This relaxation requires replacing the input simplex with the unit hypercube.

(b) Constructing $r_2$ (yellow) using $\mathbf{T}_2$ from the Anderson relaxation.

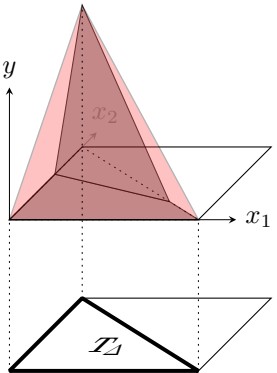

(c) Constructing $r_\triangle$ using the input simplex $\mathbf{T}_\triangle$. Our relaxation can directly model the input simplex $\mathbf{T}_\triangle$.

Figure 1: Visualisation for the intuition behind why our relaxation requires only a linear number of inequalities, whereas the Anderson relaxation [Tjandraatmadja et al., 2020] requires an exponential number of constraints. Note that since the Anderson relaxation [Tjandraatmadja et al., 2020] relaxation cannot handle the simplex constraint on the input, we have to replace the input constraint with the unit hypercube. Sub-figures (a) and (b) show the construction of the simplices in Kuhn triangulation of the unit $[0, 1]^2$ cube, which requires an exponential number of simplices. The bottom figure in each sub-figure shows the simplex and the top figure shows the constraint corresponding to the simplex. Since there are an exponential number of simplices, an exponential number of constraints are required to describe the convex hull.

Sub-figure (c) shows the input simplex $\mathbf{T}_\triangle$, and the upper bound (in red) corresponding to our relaxation. It can be noted that we only require one upper bound, and a total linear number of inequalities to describe the convex hull for the composition of a linear function with a convex activation function, when the input lies in a simplex.

It requires conditioning both layer $k$, $(h_k)$ and layer $k + 1$, $(h_{k+1})$. We first condition layer $k$. Let the cut be $\sum_i x_{k,i} \le \alpha$, where $\alpha_k > 0$. We can write it as

$$\sum_i \frac{x_{k,i}}{\alpha_k} \le 1 \tag{8a}$$

$$\sum_i \frac{\sigma(\sum_j W_{k,ij} x_{k-1,j} + b_{k,i})}{\alpha_k} \le 1 \tag{8b}$$

$$\sum_i \sigma(\sum_j \frac{W_{k,ij}}{\alpha_k} x_{k-1,j} + \frac{b_{k,i}}{\alpha_k}) \le 1 \tag{8c}$$

$$\sum_i \tilde{x}_{k,i} \le 1. \tag{8d}$$

We have achieved $\sum_i \tilde{x}_{k,i} \le 1$ by down-scaling each $x_{k,i}$ by a factor $\alpha_k$. Note that we also need to condition layer $k + 1$, $(h_{k+1})$, such that the final output remains the same. This is achieved by up-scaling the weights of layer $k + 1$ by a factor $\alpha_k$.

### 1.4 Opt-Lirpa Planet Baseline

---

**Algorithm 1** Opt-Lirpa Planet

---

1: **function** OPT-LIRPA_PLANET($\Psi$)
2:   Initialise $\underline{a}$ with values between **0** and **1**
3:   **for** $t \in [\![0, t_{max} - 1]\!]$ **do**
4:     $L^P(\underline{a}^t) = $ OPT-LIRPA_BACKWARD($\Psi, \underline{a}^t$)
5:     Compute gradients $\frac{dL^P}{d\underline{a}^t}$ via backpropagation
6:     $\underline{a}^{t+1} \leftarrow$ update gradient ascent (or Adam)
7:     $\underline{a}^{t+1} \leftarrow \pi(\underline{a}^{t+1})$      (projection)
8:   **end for**
9:   **return** $L^P(\underline{a}^{t+1})$
10: **end function**
11: **function** OPT-LIRPA_BACKWARD($\Psi, \underline{a}$)
12:   $f_N \leftarrow \Psi$
13:   **for** $k \in [\![n - 1, 0]\!]$ **do**
14:     Set $f_k(\boldsymbol{x}) \leftarrow f_{k+1}^-(\mathbf{u}_k(\boldsymbol{x})) + f_{k+1}^+(\underline{a}_k \odot \mathbb{L}_k(\boldsymbol{x})) + f_{k+1}^c$.
15:   **end for**
16:   $L^P(\underline{a}) = \min_{\boldsymbol{x}_0 \in \Delta} f_0(\boldsymbol{x}_0)$
17:   **return** $L^P(\underline{a})$
18: **end function**

---

As mentioned in the main paper, the Opt-Lirpa Planet baseline uses a similar algorithm as our proposed simplex verify algorithm, with the only difference being that it does not have the upper bound corresponding to our relaxation. More precisely, it solves the following optimization problem

$$L^P(\underline{a}) = \min_{\boldsymbol{x}} \quad \Psi(\boldsymbol{x}_{n-1}) \tag{9a}$$

$$s.t. \quad \boldsymbol{x}_0 \in \Delta \tag{9b}$$

$$\boldsymbol{x}_k \ge \underline{a}_k \odot \mathbb{L}_k(\boldsymbol{x}_{k-1}) \qquad k \in [n - 1], \tag{9c}$$

$$\boldsymbol{x}_k \le \mathbf{u}_k(\boldsymbol{x}_{k-1}) \qquad k \in [n - 1]. \tag{9d}$$

Note that there is only one upper bound, and thus there is no need for an upper weighting coefficient $\overline{a}$. The complete algorithm is shown in Algorithm 1.

## 2 Experimental Appendix

In this section, we present experimental details for the experiments presented in the main paper. We also present a comparison to other baselines.

## 2.1 Comparison to disjunctive relaxation from Anderson et al. [2020]

In Anderson et al. [2019, 2020], the authors proposed a tight relaxation that describes the convex hull of a composition of a linear function of a vector within an $\ell_\infty$ ball with the ReLU non-linearity. Although the relaxation has an exponential number of constraints, it admits a linear time separation oracle. More recently, de Palma et al. [2021] proposed an efficient dual algorithm for this relaxation. The dual algorithm uses Lagrangian relaxation, and maintains and updates a set of active constraints (Active Sets). Please see de Palma et al. [2021] for more details. In this section, we establish a comparison with the Active Set method [de Palma et al., 2021]. This relaxation does not directly handle the simplex constraint on the input. Since it relies on $\ell_\infty$ constraints on the input to derive the upper bounds, we use the unit hypercube to derive the upper bounds for the first layer. For the intermediate layers, we can use the intermediate upper and lower bounds to form the relaxation. We also compare with the Bigm-adam solver presented in [de Palma et al., 2021], which is a Lagrangian relaxation based dual solver for the unprojected version (Bigm) of the planet relaxation.

The Verification accuracy and time taken are compared to other solvers presented in the main paper, in Table 1. Note that Bigm-adam and Active Set methods also use the same intermediate bounds as the other methods. We compare the efficiency of the different methods in computing the final layer bounds. We use 850 iterations for Bigm-adam as is used in de Palma et al. [2021]. Active set method is initialized with Bigm-adam run for 500 iterations, and the active set is then run for 100 iterations. The active set uses 2 inequalities, which is the same as is used in de Palma et al. [2021].

It can be seen that our Simplex Verify achieves better verified accuracy than the Active Set solver, while being 2 orders of magnitudes faster. One main limitation of the anderson relaxation [Anderson et al., 2020] is that it requires multiple upper bounds to define the convex hull. In comparison, our method only requires a single upper bound to describe the convex hull. In future work, it would be interesting to explore a combination of upper bounds from the anderson relaxation [Anderson et al., 2020] and our proposed relaxation, for even tighter verification.

## 2.2 Experimental Setting, Hyper-parameters

Hyper-parameter tuning for $\ell_1$ perturbation experiments was done on a small subset of the CIFAR-10 test set, on the adversarially-trained Wide network. All the methods use the same intermediate bounds. The intermediate bounds were computed using Opt-Lirpa planet run for 20 iterations. The weighting coefficients are optimized using the Adam optimizer [Kingma and Ba, 2015]. The coefficients

| Dataset | | MNIST | | CIFAR-10 | | |
|---|---|---|---|---|---|---|
| Model | | Wide | Base | Wide | Deep | Wide |
| Training | | SLIDE | SLIDE | SLIDE | SLIDE | SGD |
| Accuracy | Nominal | 98.80% | 75.1% | 79.3% | 72.1% | 74.4% |
| | Pgd | 98.23% | 73.5% | 77.0% | 69.8% | 73.3% |
| Verified Accuracy | Gurobi Planet | 31.7% | 34.1% | 18.4% | 11.1% | 13.5% |
| | Gurobi Simplex | 45.2% | 48.6% | 29.4% | 13.4% | 23.7% |
| | Bigm-adam [de Palma et al., 2021] | 31.4% | 33.6% | 17.8% | 10.6% | 13.4% |
| | Active Set [de Palma et al., 2021] | 43.0% | 45.5% | 26.8% | 10.9% | 20.9% |
| | Opt-Lirpa Planet | 31.0% | 33.7% | 17.9% | 10.8% | 13.5% |
| | Simplex Verify | 44.6% | 48.0% | 28.8% | 13.4% | 22.4% |
| Avg. Verified Time/Sample | Gurobi Planet | 74.61s | 22.80s | 114.92s | 86.84s | 114.70s |
| | Gurobi Simplex | 72.47s | 22.95s | 72.17s | 59.22s | 70.42s |
| | Bigm-adam [de Palma et al., 2021] | 4.46s | 4.36s | 4.56s | 6.80s | 4.45s |
| | Active Set [de Palma et al., 2021] | 4.45s | 4.63s | 4.60s | 7.00s | 4.63s |
| | Opt-Lirpa Planet | 0.04s | 0.04s | 0.04s | 0.06s | 0.04s |
| | Simplex Verify | 0.04s | 0.04s | 0.04s | 0.05s | 0.04s |

Table 1: Verified accuracy and verification time of different solvers on MNIST and CIFAR-10 models. We test on the entire test set for MNIST, and random 1000 test images for CIFAR-10. Simplex Verify denotes our proposed solver. Our proposed method achieves much higher verified accuracy in comparison to the state of the art baseline, in the same amount of time.

| Network Name | No. of Properties | Network Architecture |
|---|---|---|
| OVAL-BASE | 1000 | Conv2d(3,8,4, stride=2, padding=1)
Conv2d(8,16,4, stride=2, padding=1)
linear layer of 100 hidden units
linear layer of 10 hidden units
(Total ReLU activation units: 3172) |
| OVAL-WIDE | 1000 | Conv2d(3,16,4, stride=2, padding=1)
Conv2d(16,32,4, stride=2, padding=1)
linear layer of 100 hidden units
linear layer of 10 hidden units
(Total ReLU activation units: 6244) |
| OVAL-DEEP | 1000 | Conv2d(3,8,4, stride=2, padding=1)
Conv2d(8,8,3, stride=1, padding=1)
Conv2d(8,8,3, stride=1, padding=1)
Conv2d(8,8,4, stride=2, padding=1)
linear layer of 100 hidden units
linear layer of 10 hidden units
(Total ReLU activation units: 6756) |
| VNN-COMP-Med | 1000 | Conv2d(3, 32, 5, stride=2, padding=2)
Conv2d(32, 128, 4, stride=2, padding=1)
linear layer of 250 hidden units
linear layer of 10 hidden units
(Total ReLU activation units: 16634) |
| VNN-COMP-Big | 1000 | Conv2d(3, 32, 3, stride=1, padding=1)
Conv2d(32, 32, 4, stride=2, padding=1)
Conv2d(32, 128, 4, stride=2, padding=1)
linear layer of 250 hidden units
linear layer of 10 hidden units
(Total ReLU activation units: 49402) |

Table 2: For each incomplete verification experiment, the network architecture used and the number of verification properties tested, a subset of the CIFAR-10 test dataset. Each layer but the last is followed by a ReLU activation function.

corresponding to the lower bounds are initialized using CROWN coefficients. The initial and final learning rates are $10^{-5}$ and 1 respectively.

We compare the efficiency of the different methods in computing the final layer bounds. For a fair comparison, the iterations for both the Lirpa style algorithms (Opt-Lirpa Planet and Simplex Verify) are tuned such that they take the same amount of time. The weighting coefficients corresponding to the lower bounds in both the methods are initialized with CROWN coefficients. Opt-Lirpa Planet is run for 6 iterations, and Simplex Verify is run for 3 iterations. The weighting coefficients are optimized using the Adam optimizer [Kingma and Ba, 2015]. The initial and final learning rates for the weighting coefficients corresponding to the lower bounds are $10^{-5}$ and 10 respectively, for Simplex Verify. The initial and final learning rates for the weighting coefficients corresponding to the upper bounds for Simplex Verify are $10^2$ and $10^3$ respectively.

Details about the network architectures used for the $\ell_1$ experiments are presented in Table 2. Note that the MNIST network uses the same architecture as CIFAR-Wide network, except that it has 1 input channel. These architectures are the same as used in de Palma et al. [2021].

The networks for multi-modal experiment are also trained with adversarial training. Both during training and verification, we allow arbitrary text perturbations from the vocabulary. For these attacks, the vocabulary comprises of the 1000 most frequent words from the training dataset. We also selected a subset of 10 classes from the Food-101 dataset. These classes include donuts, pizza, french fries, ice cream, onion rings, chicken wings, pad thai, apple pie, chicken curry, waffles.

The models in both, $\ell_1$ robustness verification and multi-modal classifier robustness verification, are trained using SLIDE (sparse $\ell_1$-descent attack) from Tramer and Boneh [2019]. Tuning of the sparsity constant for the SLIDE attack was crucial for training robust networks for the multi-modal classifier on the Food-101 dataset. We used sparsity of 0.3 for all the networks on this dataset. The

same sparsity was used for computing the upper bounds for CIFAR networks. We noted that the SLIDE attack performed much better than the normal $\ell_1$ PGD attack for training on Food-101 dataset. We also tried the EAD attack for the MNIST network, where SLIDE accuracy was $98.2\%$ and the EAD accuracy was $98.3\%$. SLIDE performs at par with the EAD attack while being computationally much more efficient than the EAD attack. See Appendix C of Tramer and Boneh [2019] for an empirical comparison between the EAD attack and SLIDE. This was the motivation for choosing the SLIDE attack over the EAD attack.