# OpenReview forum: "Overcoming the Convex Barrier for Simplex Inputs"
_NeurIPS.cc/2021/Conference — NeurIPS 2021 Poster_

### Official Review · Reviewer_vqRE · 2021-07-16

**Rating:** 6
**Confidence:** 4

**Summary:**

The paper proposes  a particular relaxation for the verification of neural network robustness under perturbation that can be phrased as functions over the probability simplex (e.g. $\ell_1$-norm).

Mathematically, the verification is first phrased as an optimization problem, that mirrors the layer structure of the neural network.
For the ReLU (or other convex activation functions) they employ the well-known convex relaxation from [Ehlers, 2017]* ("Planet") and improve upon it by a new upper bound, which utilizes the fact that the input (and all intermediate neuron values post-activation) are subject to a simplex constraint. This relaxation forms the convex hull of the possible values of a post-activation neuron and thus is tight (for a single neuron).
The resulting optimization problem forms a linear program with 2 linear lower bounds, 2 linear upper bounds and a simplex constraint for each neuron.
To allow for faster and more scalable evaluation the optimization problem is relaxed following the approach of [Zhang et al. 2018]. The 2 upper and lower bounds are relaxed into one each, that is a linear interpolation weighted by an $\alpha  \in [0, 1]$. In this formulation the optimum of the problem can be found by alternatingly computing the neuron bounds (as a function of fixed $\alpha$) and optimizing the values of $\alpha$ via gradient decent.

Experimentally the authors show that both, the relaxed version and the linear program version outperform Plant relaxations in terms of certified accuracy by a large margin on image classification (MNIST and CIFAR-10) under $\ell_{1}$ perturbations and robustness to text-perturbations for BoW-representations in an multi-modal setting (simplified Food-101). The relaxed version almost matches the linear program version in performance, but is significantly faster.

*Citations in the [Name, Year] format refer to citations in the paper.


**Limitations And Societal Impact:**

The authors do briefly consider the broader impact of their work, in a way that seems fitting given the setting of the paper.


**Main Review:**

The work seems relevant to the community and technical parts are well presented and are clear about their limitations. The mathematical arguments seem correct.
However, outside of the mathematical parts, the text and presentation of the paper seems rushed: formulations are repeated within the same argument (particularly in Section 5 or L204/207), Figure 1 makes it hard to see the difference between the bounds, and for Theorem 3.1 it may be useful to the reader to state the key idea of the proof (linear functions attain the same maxima over both $\mathcal{S}$ and $\mathcal{CH}$).

The work seems well connected with related works (with the notable exception of [1], which describes a similar algorithm to [Zhang et al. 2018b]*)

Questions:
- Can you provide further motivation for simplex constraints beyond $\ell_1$ and bag-of-words? Can you approach be used as a building block to handle self-attention layers (which usually form a probability simplex) in neural networks even if the input specification is not in the simplex constraint?
- Can you contrast your work with [2]? They also seem to provide a lifting from $\ell_{\infty}$ analysis techniques to other norms (and thereby the simplex constraint from $\ell_{1}$). (As the work is more-or-less parallel I don't expect a full numerical comparison.)
- While your focus is mostly on convex activation functions, could you also handle S-shaped activation functions like [Zhang et al. 2018b, 1]? (I assume their bound, based on the upper and lower bound pre-activation would also work for you.) Is the key issue preventing this the argument from L173?
- Am I correct in assuming that Gurobi Simplex in Section 5 utilizes the relaxation presented in 3.2?

Minor:
- L117, bad line break before the fullstop.
- L125 should likely be $\mathbf{x}_{k}, k \in [n-1]$.
- Figure 4 should be a Table.

*Citations in the [Name, Year] format refer to citations in the paper, while [Number] citations refer to:

[1] An Abstract Domain for Certifying Neural Networks, Singh et al., POPL'19

[2] Fast and Precise Certification of Transformers, Bonaert et al., PLDI'21


**Time Spent Reviewing:**

9

---

> ### Author Response · Authors · 2021-08-10
> **Reply to Reviewer vqRE**
>
> We would like to thank the reviewer for their positive assessment and feedback.
> - Through L204 we wanted to convey that Eq9 allows to make a recursive algorithm to compute bounds for $x_{k-1}$. In L207, we wanted to convey that overall Eq7 can thus be solved with a recursive algorithm. We will improve the formulation of these sentences. Fig.1 in the supplementary shows the difference in the bounds in a larger figure. We will improve Fig.1 in the main paper and add an intuition of the proof in the main text. We will also discuss the connection of our work to [1].
> - Another example beyond l1 and bag-of-words is when we have crowd-sourced noisy labels as inputs to a model, we can take the averaged one-hot label as an input feature and that would be constrained to be in the simplex. Furthermore, as hinted by the reviewer, we believe that our work can be used as a component for building relaxations for self-attention layers, but not in isolation. The output of the self-attention layer will lie in the convex hull of the value vectors.  Hence, our framework can be used as part of a tight relaxation for the self-attention layers. However, it would additionally require relaxations of other non-linearities such as the softmax function and the bilinear function.
> - Bonaert et al. propose a new convex relaxation for a number of settings of interest, but none of these characterize the convex hull exactly in any setting. On the other hand, we characterise the convex hull for our setting of a composition over a convex activation function and an activation function where the input is constrained to be in the simplex. Thus ours is the tightest possible convex relaxation for the composition of a linear function with a convex non-linearity for simplex inputs. We would be happy to add a citation to this work and a discussion of the differences in the formulation and theoretical results between our paper and Bonaert et al in the camera ready version, should the paper be accepted.
> - Yes, the reviewer is correct regarding this. The argument from L173 is crucial for our analysis. This would prevent a straightforward application to S-shaped activation functions.
> - Yes, Gurobi Simplex solves the relaxation proposed in sec 3.2 using Gurobi.
> - Minor: Thanks for pointing these out. We will fix these.

---

> > ### Comment · Reviewer_vqRE · 2021-08-31
> > **Thanks**
> >
> > Dear Authors,
> >
> > Thank you for the reply.
> > After reading the other reviews, including the discussion with aXAA, as well as your replies I retain my initial score.
> > If all discussed items are included in a revision I don't have major concerns.
> >
> > Best,
> > Reviewer vqRE

---

### Official Review · Reviewer_BsvV · 2021-07-17

**Rating:** 6
**Confidence:** 3

**Summary:**

This paper proposes to use probability simplex specifications for robustness verification. With the simplex, the convex hull for relaxing activation functions can be made tighter using a newly proposed upper bound for relaxation. Then the simplex specification and the new relaxation are used in solving the verification problem in a manner similar to backward bound propagation in CROWN, and the relaxation parameters are optimized with projected gradient ascent. By applying the method to $\ell_1$ robustness verification, and also a multi-modal setting with text perturbation, experiments show that using the simplex specification with the proposed method leads to better verified accuracy and low verification cost.

**Limitations And Societal Impact:**

They are discussed in the conclusion section.

**Main Review:**


Pros:
* The paper novelly proposes to use simplex input specification for $\ell_1$ robustness verification, which enables a tighter relaxation and upper bounds for activations.
* The proposed method leads to better verified accuracy with low computational cost on $\ell_1$ robustness verification.


Cons:
* There lacks a detailed discussion on the convex barrier which is the focus of this paper, and why the proposed method can overcome the barrier.
* The multi-modal setting (with both image and text as input) does not seem to quite make sense to me. The text can be arbitrarily perturbed, with arbitrary length and substitution words from the vocabulary. If a model should be robust to such perturbation, should it just ignore the information from the text?

Minor comments:
* Line 296, Jia et al., 2019 was published in 2019 and is probably not a concurrent work, unless there is an earlier preprint of this work.

-----

Updates after rebuttal

Thanks to the authors for the rebuttal! With the rebuttal, I think now I can better understand how the convex barrier is overcome. I think it will be good if the authors can include more detailed discussions on the convex barrier and reason that the proposed method can overcome the barrier into the revision, since currently "barrier" does not appear again in the main text since page 2, which makes it difficult to understand.

For Sec 5.2 on the multi-modal experiments, I still think it seems to be unreasonable to me, because the model can trivially become robust to such perturbations by totally ignoring the actually useless text. I think it may be better to use a more reasonable setting to demonstrate the benefits of the proposed method.

Overall, I'm currently on the borderline for this paper.


**Time Spent Reviewing:**

5

---

> ### Author Response · Authors · 2021-08-10
> **Reply to Reviewer BsvV**
>
> We would like to thank the reviewer for their valuable feedback.
> - Convex relaxation barrier is a term introduced in [Salman et al, 2019]. The feasible set of the verification problem is non-convex, so various methods use a convex relaxation of the actual problem. Convex barrier is the gap between the optimal value of the original verification problem and the optimal convex relaxation of the non-linearity. We propose the optimal convex relaxation [Theorem 3.1] for the composition of a linear layer with a convex non-linearity, when the input lies in a simplex. We further show that the proposed relaxation is tighter than the optimal convex relaxation for just the non-linearity [Proposition 3.2]. Since our relaxation is tighter than Planet, we overcome the convex barrier.
> - The multi-modal experiment is motivated from models trained when text is scraped from the web, as is done in Food-101 data collection. This is a common scenario as there are large amounts of data in this format on the web and social media platforms. The point of using text is to improve the performance over just the image modality. But this makes the model vulnerable to noise in the text, i.e., incorrect text making the model mis-classify despite the correct image. There is a trade-off here, and ideally the model should have higher nominal accuracy than the accuracy of the image-only model, while not being too vulnerable to noise in the text. The aim of this specification is to characterise the worst-case sensitivity of the model. We are not aiming for perfect robustness to the noise in text, but aim to check its sensitivity. To this end, we provide results for networks trained with different weighting for nominal and PGD loss during the adversarial training, in Figure 4 in the submission.
> - Minor comments: We meant that [Jia et al., 2019] was concurrent to [Huang et al., 2019]. We will improve the formulation of the sentence. Thanks for pointing it out.

---

### Official Review · Reviewer_aXAA · 2021-07-20

**Rating:** 6
**Confidence:** 5

**Summary:**

The paper proposes a new convex relaxation for verifying neural networks with convex activations with specifications defined over simplex inputs. These convex relaxations are more precise than the triangle relaxation from Ehlers et. al and based on computing the convex hull of the set {(x,f(w.x+b))} where x belongs to a simplex and f is a convex activation function. The size of the resulting relaxation is linear in the number of neurons. Building on this, the authors develop a verification algorithm that propagates simplex input through the network and a solver for solving the verification problem posed as an optimization problem based on the obtained convex relaxations. The experimental results show that the verifier developed in this work is more effective than prior work.

**Limitations And Societal Impact:**

The authors address both adequately.

**Main Review:**

Originality: I found the technical contributions presented here to be incremental over prior work. The single neuron convex relaxation is based upon the framework of Tjandraatmadja et al., except that the difference here is that the input x is defined over the neurons in all previous layers and the set describing x is a simplex. Similarly, the efficient solver presented in Section 4.2 appears to be an extension of Xu et. al adapted to the simplex.

Quality: I have questions regarding the proof of Theorem 3.1 presented in the appendix. Why should a convex activation function applied on a convex set produce a convex output as mentioned in Lines 8-9 in the appendix. Further, line 12-13 says that the optimum value of a linear function over the non-convex set S is the same as on its convex hull, I am not sure if that is a true statement in general.

Eq. 4(b), of the form y>=\sigma(w.x+b), why should this always describe a convex region for all convex activations?

In lines 172-173, it is written that " If this is not the case, we can simply add a constant to the activation of each layer so that the output becomes non-negative",  wouldn't your analysis become unsound?

In lines 239-240, it says "For a target label, the image is said to be verified if the lower bound for a label is positive", so you do not verify that the network is not robust but only that it cannot be classified to a target class?

In terms of the experiments, can you compare against a state-of-the-art incomplete verifier? e.g., CROWN or DeepPoly?

Clarity: The paper motivates the problem well. However, some of the technical details might be inaccessible to a general audience. The authors should consider providing more examples to demonstrate how their method works. Also, the technical delta over the works of Tjandraatmadja et al. and Xu et. al is not clear.

Significance: Two years ago, a convex barrier was proposed which exposed the limitations of existing convex relaxations. Since then, a number of methods have been developed that provide more precise convex relaxations than the convex barrier while maintaining scalability. This work explores the same direction. The experimental results demonstrate that the proposed relaxations are effective. The global robustness property considered here can serve as a good benchmark for comparing verifiers.

Additional question:
1. Can your framework be used for computing multi-neuron relaxations as in [1]?

Overall, given the concerns above I think the paper is below the bar of acceptance, but I will be happy to increase my grades if the authors can address my concerns.


References:
1. Beyond the Single Neuron Convex Barrier for Neural Network Certification. NeurIPS 2019.

===============
Post rebuttal: The authors addressed my concerns regarding the correctness of the theorem and therefore I am happy to raise my score. However, I still find the work to be somewhat incremental over Tjandraatmadja et al. therefore I will not push for acceptance but also not strongly against seeing this paper getting accepted.

**Time Spent Reviewing:**

6

---

> ### Author Response · Authors · 2021-08-10
> **Reply to Reviewer aXAA**
>
> We thank the reviewer for the valuable feedback.
>
> #### Originality:
> We respectfully disagree regarding the lack of originality compared to prior work. Directly applying the framework of Tjandraatmadja et al. would require replacing the simplex with a unit cube, since their work requires a bound-constrained input domain. Furthermore, doing so and applying the relaxation of Tjandraatmadja et al. would lead to a formulation that involves an exponential (in the network size) number of constraints, and requiring custom cutting plane solvers and significant effort to implement in a scalable manner.
>
> In contrast, our relaxation is concise as it only has a linear number of inequalities (3 linear inequalities per neuron for ReLU activation). This allows for straightforward adaptations of algorithms like CROWN and the work of Xu et al, to our setting and leads to efficient solvers that can compute verified bounds in time comparable to the PLANET relaxation of Ehlers et al. Further:
> - Our relaxation is derived in a different way from Tjandraatmadja et al. While the proof of Tjandraatmadja et al. uses results from Submodularity and Convex Geometry, our proof uses techniques from Convex Analysis. The only thing being common is the theme, that is, a tight relaxation for the composition of linear layer and a convex nonlinearity.
> - The idea of propagating simplices through the hidden layers of the network is novel. The previous work does not leverage the input simplex, and only propagates box constraints after the input layer. We show that simplex propagation can be useful.
> - As a consequence of our proposed relaxation, Simplex-Verify improves the state-of-art for l1 robustness verification by significant margins. It also allows verifying a challenging property for the multi-modal setting. We believe that this is significant for the community.
>
> We leverage the technique of [Xu et al.] for the efficient solver in Sec 4.2. This is not our primary contribution, and indeed is a straightforward adaptation of their algorithm. However, such a straighforward adaptation is possible due to the concise nature of our convex relaxation, as opposed to that of Tjandraatmadja et al.
>
> #### Quality:
> - We would like to clarify that $\mathcal{S} = \\{ (y, \mathbf{x}) \mid y=\sigma(\mathbf{w}^T\mathbf{x} + b), \mathbf{x} \in \Delta_m \\}$, where $\sigma$ is a convex activation function. $\mathcal{S}$ is not necessarily convex. We aim to characterise the convex hull of $\mathcal{S}$, which is denoted by $\mathcal{CH}$. This is the smallest convex set that contains $\mathcal{S}$. The convex hull is characterized by a system of linear inequality constraints (equations (1a) and (1b) in the paper).
> - ''Further, line 12-13 says that the optimum value of a linear function over the non-convex set S is the same as on its convex hull, I am not sure if that is a true statement in general'': See Theorem 32.2 in Convex Analysis, R. Tyrrell Rockafellar, 1970, for a proof. We also provide an alternate proof below.
> The convex hull of set $\mathcal{S}$ is the set of all convex combinations of points in $\mathcal{S}$. Thus any point ($z$) in the convex hull can be written as a convex combination of points ($x_i$) in $\mathcal{S}$ ($z = \sum_i \lambda_i x_i$, where $\sum_i \lambda_i = 1$ and $\lambda_i \geq 0$). By the definition of convex function, value at any such point $z$ is less than or equal to the convex combination of values at those points, which in turn is less than or equal to the maximum over the set $\mathcal{S}$. And since the convex hull contains the set $\mathcal{S}$, the maximum value is actually attained. Thus the optimum value of any convex function over the non-convex set $\mathcal{S}$ is the same as on its convex hull.
> - ''Eq. 4(b), of the form $y>=\sigma(w.x+b)$, why should this always describe a convex region for all convex activations?'': $\sigma(w.x+b)$ is a convex function. Since the line segment between any two points on the graph of the function lies above the graph between the two points, this means that this line segment lies within the set $y>=\sigma(w.x+b)$, thus this set is convex.
> - L172-173: Let $\mathcal{S} + v = \\{s+v: s \in \mathcal{S}\\}$. We characterise the ConvexHull($\mathcal{S} + v$) based on our theorem. Then the ConvexHull($\mathcal{S}) = $ ConvexHull($\mathcal{S} + v$) $-v$
> - L239-240: We compare the verified accuracy, which requires us to compute the robustness margin for all possible labels. An image is said to be verified if the lower bound across all possible labels is positive.
> - Comparison to CROWN or DeepPoly: The ''Opt-Lirpa Planet'' baseline is an optimized version of CROWN, which is documented to be a strict improvement over CROWN.
>
> #### Clarity:
> We are happy to provide examples and include any suggestions from the reviewer to demonstrate the working of the method in the paper. We have discussed the relation to Tjandraatmadja et al. and Xu et. al, in the Originality comment above.
>
> #### Additional question: Multi-neuron relaxations:
> We will investigate this further in future work. Thanks for the suggestion.

---

> > ### Comment · Reviewer_aXAA · 2021-08-21
> > **Proof of Theorem 3.1 still not clear**
> >
> > Dear Authors,
> >
> > Thanks for the response. I am not sure if I follow your arguments for the proof of 3.1 in the response. Let's consider a ReLU based network with convex inputs and oi, oj be two output labels. The exact network output wrt input set is non-convex (disjunction of polyhedra). Your theorem implies that if one optimizes oi-oj over the exact non-convex set, it has the same output as the convex hull of the exact set, implying that the MILP formulation of ReLU networks has the same precision as the strongest LP formulation. This does not sound intuitive to me, am I missing something?

---

> > > ### Author Response · Authors · 2021-08-23
> > > **Clarification on MIP-LP relationship**
> > >
> > > Dear reviewer,
> > > That is indeed the case. Just to clarify, consider a network where the input is constrained to lie in a bounded convex set $X$ and we consider the set of outputs $Y=${$(o_i, o_j): o=f(x), x \in X$} and $f$ denotes the network mapping. Let $\mathcal{CH}(Y)$ denote the convex hull of the set $Y$.
> > >
> > > Then, it is indeed the case that if one solves the optimization problem
> > >
> > > $$\min_{(o_i, o_j)} o_i - o_j \text{ subject to } (o_i, o_j) \in \mathcal{CH}(Y)$$ - (1)
> > >
> > > the optimal value coincides with
> > >
> > > $$\min_{(o_i, o_j)} o_i - o_j \text{ subject to } (o_i, o_j) \in Y$$ - (2).
> > >
> > > The reason is simply that the set $\mathcal{CH}(Y)$ is a bounded convex polyhedron and there exists an extreme point of the polyhedron that attains the optimal value. By definition of the convex hull, every extreme point of of $\mathcal{CH}(Y)$ belongs to $Y$, and hence the optimal value over $\mathcal{CH}(Y)$ is attained at a point in $Y$, implying that the optimal values of problems (1) and (2) coincide.
> > >
> > > However, in practice, the description of the exact convex hull may involve an exponential number of linear inequality constraints, making problem (1) intractable even if it is a linear program. That is why, even though every mixed-integer linear program with a bounded feasible set has an exact reformulation as a linear program, that reformulation is seldom practical since the number of constraints involved in the reformulation may be exponential in the number of decision variables.
> > >
> > > The contribution of our paper is to show that when set $X$ is the simplex and set $Y$ is defined as {$(o, x): o=\text{ReLU}(w^T x + b), x \in X$}, we can characterize $\mathcal{CH}(Y)$ using a compact characterization, as described in theorem 3.1. Thus, we obtain an exact chaterization of the convex hull of the input-output set of a single neuron formed by composing a linear operation (like convolution or a fully connected layer) with the ReLU function. This would allow us to solve any problem requiring minimzing/maximzing a function of the form $o + c^T x$ for any cost vector $c$ exactly, by solving a linear program with variables $o, x$ and constraints given by those from theorem 3.1.
> > >
> > > This is the exact analog of the result from Tjandraatmadja et al., where they characterize the convex hull when $X$ is defined by box constraints rather than the simplex. The advantage of our characterization is that, when the inputs are constrained to lie in the simplex rather than in a box constrained set, the description of the convex hull only requires 3 linear inequalities per neuron, rather than the exponential number required in the formulation from Tjandraatmadja et al.

---

> > > > ### Author Response · Authors · 2021-08-23
> > > > **Connection to fundamental theorem of MILP**
> > > >
> > > > Dear reviewer, Many thanks for raising the clarification question. We would also like to point out that the equivalence of the optimal value of any mixed integer linear program (MILP) defined using rational constraint matrices and vectors and the optimal value of the LP that replaces the feasible set of the MILP with its convex hull is a direct consequence of Meyer's fundamental theorem of MILP.
> > > >
> > > > This observation has been pointed out in various surveys/books on MILP, e.g. by Conforti et al. (available at https://www.andrew.cmu.edu/user/gc0v/webpub/IPsurveyAussois-11-08.pdf). Please see the end of page 7, where it is stated: "Meyer’s theorem is the theoretical underpinning of the polyhedral approach to integer programming. Indeed, it shows that the problem of optimizing a linear function over a mixed integer set S is equivalent to solving a linear program." The survey also provides the exact statement of the fundamental theorem of MILP and its proof (please see Theorem 2.19 on page 16).

---

> > > > > ### Comment · Reviewer_aXAA · 2021-08-23
> > > > > **Thanks for the clarification and further question**
> > > > >
> > > > > Dear Authors,
> > > > >
> > > > > Thanks a lot for the clarification. It resolved some of my doubts. I have a few follow-up questions based on your response above.
> > > > >
> > > > > 1. Is the theorem valid for rational coefficients in w and b only or also with real coefficients?
> > > > >
> > > > > 2. One will not get MILP for other convex activations like Softplus, will the result hold there?
> > > > >
> > > > > 3. It seems to me that the main delta over  Tjandraatmadja et al. is the closed-form characterization of the convex hull for a single neuron when the input is simplex instead of the box constraints?

---

> > > > > > ### Author Response · Authors · 2021-08-23
> > > > > > **Clarifications**
> > > > > >
> > > > > > Dear reviewer,
> > > > > > Just to clarify, our proof of theorem 3.1 does not rely on the MILP-LP equivalence result we cited above. We quoted that just to show the general way a MILP is reformulated as an LP, in response to your earlier query. Our proof relies on the more general result from convex analysis that the optimum of a linear function over a bounded set $S$ coincides with the optimum over the convex hull of the set $\mathcal{CH}(S)$ - see slide 83 in this deck https://web.mit.edu/dimitrib/www/Convex_Slides_2014.pdf for a proof of this result.
> > > > > >
> > > > > > Regarding your specific queries:
> > > > > > 1. Since our proof relies on the general result quoted above, it does not rely on rationality of w, b and applies even when w, b are real valued.
> > > > > >
> > > > > > 2. Our result still holds for convex activations (since it follows from the general convex analysis result above). However, the resulting convex hull is not defined by only linear constraints, but by the more general set of convex constraints:
> > > > > > $$o \geq \sigma(w^T x + b) $$ (nonlinear but convex constraint)
> > > > > > $$o \leq \sum_i x_i (\sigma(w^T e^i + b) - \sigma (b)) + \sigma(b) $$ (linear constraint)
> > > > > >
> > > > > > 3. That is accurate. Moreover, as we stated earlier, in our case the convex hull is composed of 3 linear constraints (for a ReLU activation) or one convex nonlinear and one linear constraint for general convex activation functions (as described in our response to Question 2 above), as opposed to the result from Tjandraatmadja et al. which requires an exponential number of constraints. Furthermore, this simple characterisation of the convex hull allows us to derive an efficient verification algorithm based on the Algorithm from Xu et al, which is not possible in the work of  Tjandraatmadja et al., where specialized cutting plane solvers are required to handle the exponential number of constraints.
> > > > > >
> > > > > > We are happy to clarify any additional questions.
> > > > > >
> > > > > > Regards,
> > > > > > Authors

---

> > > > > > > ### Author Response · Authors · 2021-08-26
> > > > > > > **Proof that optimizing over a set = Optimizing over its convex hull**
> > > > > > >
> > > > > > > Dear reviewer,
> > > > > > > Thank you again for your questions and for engaging in a discussion around this paper.
> > > > > > >
> > > > > > > Here is a reference that actually proves the result that the optimum of a linear function over a compact set coincides with the optimum over the convex hull of the compact set:
> > > > > > >
> > > > > > > https://www.princeton.edu/~aaa/Public/Teaching/ORF523/S16/ORF523_S16_Lec4_gh.pdf (theorem 8).
> > > > > > >
> > > > > > > We hope this clarifies any concerns regarding the validity of our proof of theorem 3.1. We would be happy to clarify any additional questions you may have.
> > > > > > >
> > > > > > > Regards,
> > > > > > > Authors

---

> > > > > > > > ### Comment · Reviewer_aXAA · 2021-08-31
> > > > > > > > **Thanks for the pointers**
> > > > > > > >
> > > > > > > > Dear Authors,
> > > > > > > >
> > > > > > > > Thanks for providing more details regarding the proof, it resolves my concerns regarding the correctness of the theorem and I am happy to raise my score.

---

### Official Review · Reviewer_3E9J · 2021-07-23

**Rating:** 7
**Confidence:** 5

**Summary:**

This paper proposes an efficient bound propagation method for verifying the robustness of neural networks under L1 (simplex) constraints. The authors show that for the L1 norm setting, the convex hull for a linear and relu layer can be exactly computed efficiently, unlike the setting for Linf norm. Then, the authors designed an efficient algorithm that uses a single pass of backward bound propagation to solve the resulting linear programming problem. The bound propagation algorithm has optimizable parameters that can be optimized to achieve the tightest bound.

**Limitations And Societal Impact:**

No potential negative societal impact.

**Main Review:**

Strengths:

1. The paper has an exciting theoretical finding that when L_1 norm perturbation is used on input x, the convex hull of a relu and a linear layer can be exactly and efficiently computed. Unlike the case where the input x has a Linf norm perturbation (Tjandraatmadja et al., 2020), in the L_1 norm case there is only one simple linear upper hyperplane rather than potentially exponentially many in the number of neurons. This is a quite important finding showing that the difficulty of getting tight convex relaxations for different norms might be quite different, and may inspire future research on investigating additional norm constraints.

2. With the help of the simple and tight convex relaxation, the authors developed an efficient bound propagation based algorithm in the same spirit of CROWN (Zhang et al., 2018). This allows us to compute verification bounds very efficiently and also makes it possible to use GPU accelerators and scale to large networks. Additionally, like the optimized CROWN bounds in (Xu et al., 2021), the proposed bound propagation procedure can also be optimized to achieve the tightest possible relaxation. This is very useful for obtaining tight bounds.

3. Experimentally, the proposed approach achieves much better L_1 norm verified accuracy compared to existing Planet relaxation based verifiers such as Optimized CROWN/LiRPA, and much faster than Active Set based verifiers.


Improvements that can be made:

1. The simplex propagation in section 4.1 can produce loose bounds when the network is deep. It looks like an IBP-like bound for the L1 norm, which is exact for one layer but can become increasingly loose when propagating to later layers. Why not use the efficient solver (Section 4.2) to solve each layer's simplex constraints $\alpha_k$ by treating $1^T h_k(x)$ as the objective? This was also done in a recursive manner in the original CROWN algorithm - the intermediate layer bounds are also computed via CROWN, rather than IBP. This also enables joint optimization of intermediate layer's bounds (here it is L1 norm simplex) in a similar manner as (Xu et al., 2021), which can potentially significantly improve the results.

2. In experiments, it seems both Opt-LiRPA and Simplex Verify are very fast on the MNIST and CIFAR-10 models under testing. Ideally, to better demonstrate the scalability of the proposed method, a few larger networks should be included into evaluation (e.g., you can look for appropriate networks in VNN COMP).

3. The epsilon used in experiments are fairly small, and PGD attack accuracy is very close to the nominal accuracy. For evaluating the attack accuracy in the L1 norm setting, you can try the EAD attack [1] which is specifically designed for the L1 norm case. This can probably reduce the gap between attack accuracy and verified accuracy.

4. I think theorem 3.1 is the most important result of this paper, and it might be worthwhile to have more discussions on this and give some insights on why adding the simplex constraints on x allows an efficient relaxation, and why an efficient relaxation is not possible for Linf norm case.

Minor questions:

1. For Opt-LiRPA are the intermediate layer bounds jointly optimized like in (Xu et al., 2021)? Based on the algorithm listed in the appendix it seems not?

Overall, I feel this paper makes a significant contribution to the neural network verification problem, and I support the acceptance of this paper. However I hope the authors can further work on the possible improvements mentioned above to make this paper more solid.

References:

[1] Chen, Pin-Yu, Yash Sharma, Huan Zhang, Jinfeng Yi, and Cho-Jui Hsieh. "Ead: elastic-net attacks to deep neural networks via adversarial examples." In Thirty-second AAAI conference on artificial intelligence. 2018.



**Time Spent Reviewing:**

5

---

> ### Author Response · Authors · 2021-08-10
> **Reply to Reviewer**
>
> We would like to thank the reviewer for their positive comments and feedback.
> - We present results on Cifar10\_8\_255 and Cifar10\_2\_255 networks from the GGN-CNN benchmark in VNN comp, with epsilon 0.5 and 0.3 respectively. The Cifar10\_8\_255 network has 16634 ReLUs and Cifar10\_2\_255 has 49402 ReLUs. These results show that similar to the results presented in the submission, our approach significantly outperforms the baselines even on the larger networks.
>
> |    	| Cifar10\_8\_255  | Cifar10\_2\_255  |
> |:---------- 	|:----------|:----------|
> | Nom acc    	| 81.4\%    | 82.9\%    |
> | Pgd acc    	| 80.5\%    | 81.9\%    |
> | Opt-Lirpa Planet    | 48.8\%    | 14.1\%    |
> | Simplex-Verify    | 59.4\%    | 20.6\%    |
> | Time: Opt-Lirpa Planet    | 0.05    | 0.06    |
> | Time: Simplex-Verify    | 0.05    | 0.06    |
>
> - We ran the EAD attack on the MNIST wide network and obtained similar results to the sparse PGD attack reported in the paper. Specifically, the Sparse PGD attack accuracy is 98.2\% and the EAD accuracy is 98.3\%. We will try it for CIFAR networks too.
>
> - We discussed the intuition behind why adding simplex constraints on the input allows an efficient relaxation in Appendix 1.2 under 'Linear vs exponential'.
> The relation can be understood using the Kuhn-triangulation of $[0,1]^n$ [Todd, 1976]. This is used in the submodularity based proof in [Tjandraatmadja et al., 2020]. The Kuhn triangulation is used to describe the collection of simplices whose union is $[0,1]^n$. It requires an exponential number of simplices to describe the unit hypercube. A unique affine interpolation needs to be constructed on each of these simplices. This gives an overall exponential number of inequalities. In contrast, our relaxation only requires a linear number of inequalities. It is also important to note that the input simplex $\Delta_n$ is not one of the simplices from the Kuhn triangulation whose union is the unit hypercube (see Figure 3 and Example 1 in Appendix of Tjandraatmadja et al. [2020] for an instance).
>
> - We will try our solver from Section 4.2 for the simplex propagation. Thanks for the suggestion.
> - Minor question: The intermediate bounds are optimized layer-wise, just like Simplex-Verify.

---

> > ### Comment · Reviewer_3E9J · 2021-08-27
> > **Thank you for the response.**
> >
> > I thank the author for the response. The results on VNN COMP models look promising and thank you for providing more explanations for the relaxations. In fact, I feel the main idea of the paper was not clearly presented. The intuitions of simplex relaxation as you explained to me and other reviewers should be an important part of the paper, and it is also better to clearly show the differences between this relaxation and the ones in (Tjandraatmadja et al., 2020) in the main text, using some concrete examples. I feel the issues from other reviewers are mostly because the main idea and intuitions of this paper are not well presented.
> >
> > A few things I hope the authors can include in the final version:
> >
> > 1. use a recursive simplex propagation for intermediate layer bounds (just like in CROWN) as mentioned in Improvements (1) in my review
> > 2. include EAD attack results into experiments.
> > 3. make sure to mention that in Opt-LiRPA the intermediate layer bounds are not jointly optimized.
> > 4. include more discussions on intuitions of the relaxation, and include a low dimension example, preferably using figures.
> >
> > Overall, I feel the paper does make a significant contribution, although its current version is preliminary and does have a lot of room for improvements. I support acceptance of this paper, but the author should also address the issues with other reviewers.

---

### Decision · Program_Chairs · 2021-09-27

**Decision:**

Accept (Poster)

**Comment:**

The paper studies the robustness verification problem for neural networks over simplex inputs. Unlike the Linf case where a tight bound requires exponential complexity, the paper proposes an efficient method to propagate the simplex through ReLU networks where the size of relaxation remains linear in the number of neurons and shows the algorithm can successfully compute tight bounds on several benchmarking datasets. After discussions, the reviewers think that although the method is an extension of (Tjandraatmadja et al.), the derived closed-form expression is new and it is interesting to see that such a nice form can be obtained in the L1 case. Therefore we recommend acceptance of the paper.

The reviewers also think the paper is not well written and hope the authors improve the presentation of the paper based on the review comments.